# Self-Attention Augmentation with Smoothed Noise Injection for Enhancing Transformer Fine-Tuning on Temporally Structured Health Data

## Abstract

Pre-training Transformer models on self-supervised tasks and fine-tuning them on downstream tasks, even with limited labeled samples, have achieved state-of-the-art performance across various domains. However, learning effective representations from complex temporal structured health data and fine-tuning for clinical risk predictions remains challenging. While self-attention mechanisms are powerful for capturing relationships within sequences, they can struggle to model intricate dependencies in event sequences, especially when training data is limited. Existing solutions often rely on expensive modifications to the pre-training phase. In this work, we propose a novel method, Smoothed Noise Injection Self-attention Augmentation (SNSA), to augment Transformer models during the training. Our approach encourages the self-attention mechanism to effectively learn complex dependencies within input sequences. This is achieved by introducing noise to the self-attention and then smoothing it via convolving with a 2D Gaussian kernel. The first term perturbs the attention between events , encouraging the model to explore diverse attention patterns. The Gaussian smoothing adaptively filters this noise, allowing the model to focus on more relevant events within each sequence. With SNSA, we observe enhanced model performance on downstream tasks. Furthermore, our method sheds light on the model's ability to learn complex relations within a sequence of medical events, providing valuable insights into its behavior within the attention mechanism.

## 1 Introduction

Foundation models, deep neural networks pre-trained on broad unlabeled data using self-supervised methods, have significantly impacted various aspects of our lives, including law, healthcare, education, and more Bommasani et al. (2021); Guo et al. (2023); Wornow et al. (2023). These models typically acquire general knowledge about the data through pre-training a variant of the Transformer network on a self-supervised task like Masked Language Model (MLM), and then adapt this knowledge to downstream tasks with only a few labeled samples during the fine-tuning process. Researchers showed that pre-training, even with limited data, can improve Transformers' performance significantly Amos et al. (2023).

Pre-training Transformers have been employed with various self-supervised objectives and domains. Common objectives include corrupted text reconstruction tasks like MLM Devlin et al. (2018); Lewis et al. (2019); Lan et al. (2019) and standard language models such as next-word prediction Radford et al. (2019); Brown et al. (2020), which have been extensively utilized Liu et al. (2023). These models typically adopt a backbone architecture inspired by the multi-head attention mechanism in Transformers Vaswani et al. (2017), known for its effectiveness in modeling complex interaction between events (tokens) in a sequence (text). These foundation models have been pre-trained on different domain data Lan et al. (2019); Radford et al. (2019), including structured temporal health data as sequences of events Li et al. (2020); Rasmy et al. (2021); Pang et al. (2021).

Modeling Electronic Health Records (EHRs) trajectories presents a critical opportunity for predicting health-related outcomes, offering benefits like early intervention, cost reduction, and improved public health. This

field has attracted significant attention from deep learning researchers Xiao et al. (2018); Amirahmadi et al. (2023); Boll et al. (2024). Typically, healthcare specific foundation models are pre-trained on publicly available, unlabeled EHR data, and adapting these models through fine-tuning consistently demonstrates superior performance across various tasks Li et al. (2020); Rasmy et al. (2021); Pang et al. (2021); Ren et al. (2021).

However, EHRs are often scarce, and training Transformers to learn the complex relationships between medical events in longitudinal EHRs requires either large amounts of data, or advanced training techniques and augmentations Dosovitskiy et al. (2020); Touvron et al. (2021); Hassani et al. (2021; 2023). Due to privacy concerns and the scarcity of publicly available datasets, models often fail to learn the intricate dependencies between events in a patient's history. To address this, (Choi et al., 2020) proposed incorporating domain knowledge into the attention mechanism, while (Zhu & Razavian, 2021) employed variational regularization. Additionally, (Amirahmadi et al., 2024) suggested pre-training the Transformer on the MLM task and the ordering of medical events in a patient's history, and (Kim & Lee, 2024) proposed using learnable, adaptive kernels in the attention matrices to improve contextual representations and enhance the learned structure through self-attention. Figures 1 and 4 illustrates how these various approaches impact self-attention behaviors in leaning the relations between events. However, these methods often come with substantial computational costs and require extra effort for implementation and design.

Data augmentation is another solution to tackle the data scarcity challenge. Augmenting data with discrete data types, such as series of medical codes or tokens in text, is challenging because small perturbations can drastically alter semantic meaning, and interpolation in discrete space is not feasible Chen et al. (2020). As a result, researchers have proposed augmenting models during training as an alternative Jain et al. (2023); Zehui et al. (2019); Wu et al. (2023).

In this study, we propose a simple two-step augmentation technique—Smoothed Noise Injection Self-Attention (SNSA)—that perturbs attention scores by injecting adaptive Gaussian noise followed by smoothing with a Gaussian kernel. Our investigation of attention distributions reveals that fine-tuned Transformers tend to produce highly polarized attention scores—values clustering near the extremes (0 or 1), which restricts the model's capacity to explore diverse dependencies (see the bottom row of Figure 4). By introducing controlled noise into attention scores during fine-tuning, we encourage exploration of alternative dependency paths between events. The subsequent smoothing operation helps restore structural consistency while preserving diversity, resulting in more balanced and informative self-attention maps.

The main contributions are summarized as follows:

1. We proposed a simple self-attention augmentation method that encourages the model to explore and learn more complex attention patterns during fine-tuning. Importantly, this approach does not modify the computational graph, making it easily applicable to any pre-trained Transformer.

2. We conducted several evaluations on various downstream tasks, examining the effect of the novel method on model performance, model robustness with limited training samples,and the balance of attention distribution between distant and nearby events. Our results demonstrate how it improves the performance of pre-trained Transformers.

## 2 Preliminary

### 2.1 Transformer encoder and self-attention

The core back-bone of Transformers encoder is the multi-head self-attention. Each self-attention head is:

$$Q_h = XW_h^Q, K_h = XW_h^K, V_h = XW_h^V, \tag{1}$$

$$A_h = \text{softmax}(\frac{Q_h K_h^T}{\sqrt{d_k}}) \tag{2}$$

$$H_h = \text{Self-attention}(X) = A_h V_h \tag{3}$$

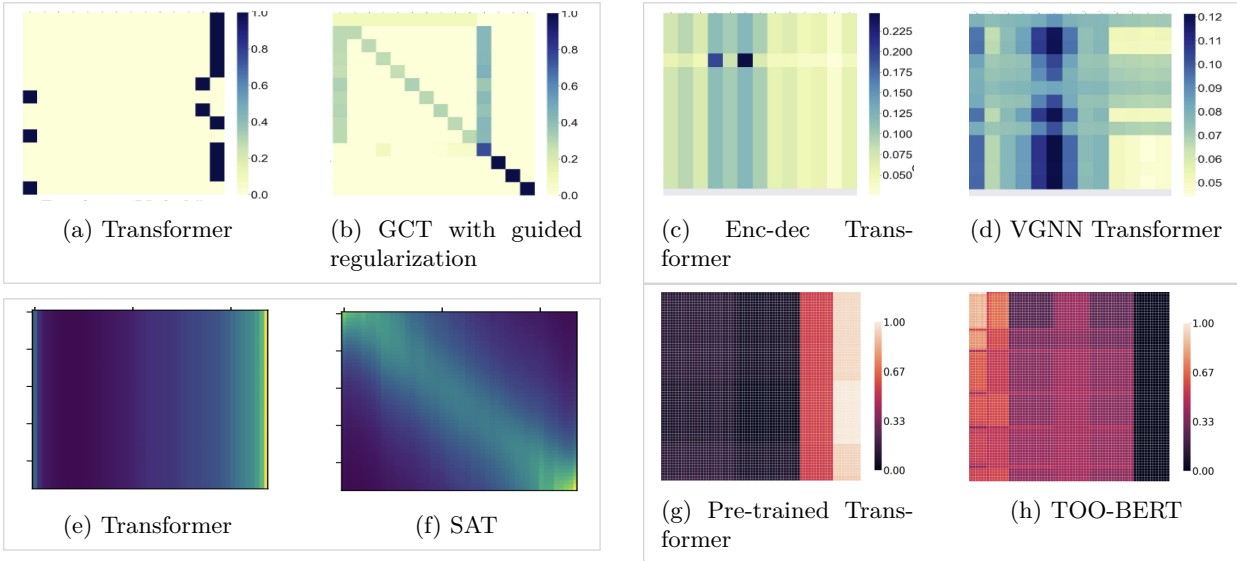

Figure 1: Visualization of attention score patterns for different models from previous studies and how their proposed methods helping a more complicated structure in attention scores in Transformers. (a–b) Transformer trained from random weights vs. Transformer trained with domain knowledge (Zhu & Razavian, 2021; Choi et al., 2020). (c–d) Encoder-decoder vs. VGNN using variational regularization (Zhu & Razavian, 2021). (e–f) Vanilla Transformer vs. SAT with temporal priors (Kim & Lee, 2024). (g–h) Transformer pre-trained on MLM vs. MLM with trajectory order prediction (Amirahmadi et al., 2024). Panels (e–f) had no color bars in the original papers.

Where, $Q, K \in \mathbb{R}^{n \times d_k}$ and $V \in \mathbb{R}^{n \times d_v}$ and $n$ is the length of input sequence and $d_k$ and $d_v$ are dimenssion of Key and Value. $A_h$ is the attention score matrix and each $A_{i,j}$ indicates how much attention token $x_i$ put on $x_j$. Transformer encoders, is built on concatenation of $| h |$ number attention heads in parallel, so each one has its own weights. Then, the concatenation is projected:

$$\text{MultiHead}(X) = \text{Concat}(H_1...., H_{|h|})W^O \tag{4}$$

Where, $W^O \in \mathbb{R}^{|h| \times d_v}$ Multiple self-attention heads in parallel, help the model to attend to information from different representation subspaces (Vaswani et al., 2017; Hao et al., 2021).

## 2.2 Pre-training, fine-tuning

Pretraining typically involves the model acquiring general knowledge, which is then used to initialize the final network. Subsequently, the final network adjusts these weights to obtain optimized weights for specific downstream tasks Chen et al. (2021). This approach has been extensively utilized for adapting foundation models to downstream tasks Lan et al. (2019); Liu et al. (2023).

## 3 Related works

Advanced training techniques and data augmentation have been widely adopted to improve the performance of Transformer models, especially in settings with limited labeled data. These methods aim to enhance the generalizability and robustness of learned representations.

Several methods modify self-attention to better learn intricate local and global attentions between different tokens. Hassani et al. (2023) introduced a sliding window attention mechanism to localize attention spans and improve efficiency. Ding et al. (2023) reduced attention complexity by segmenting key, query, and value inputs and sparsifying their interactions, allowing Transformers to better model both short- and long-range dependencies. Positional encoding has also been a target for improvement: Su et al. (2024) and Press

et al. (2021) enhanced distant token interaction by encoding absolute positions with rotation matrices or distance-based penalties on query-key attention scores. While these methods are effective, they often require structural changes to the attention mechanism, making them less compatible with pre-trained models and harder to integrate into existing pipelines.

Data augmentation is another solution to tackle the data scarcity challenge, but it is particularly challenging in discrete domains like medical codes or text, where small changes can drastically alter semantic meaning and interpolation is not well-defined (Chen et al., 2020). To address this, researchers have proposed augmenting models during training or fine-tuning by injecting noise into internal representations (Jain et al., 2023; Zehui et al., 2019; Yuan et al., 2022; Wornow et al., 2023; Wu et al., 2023). Injecting Gaussian noise into activations has been shown to help models converge to smoother minima, improving generalization, calibration, and robustness to perturbations (Camuto et al., 2020). Zhu et al. (2019) enhanced the performance of BERT (Devlin et al., 2018) and RoBERTa (Liu et al., 2019) by adding adversarial noise to word embeddings, a technique later extended to graph neural networks by Kong et al. (2022) for improved out-of-distribution generalization. In the self-attention space, Zehui et al. (2019) proposed DropAttention, which randomly masks and expands attention scores to regularize focus. Similarly, Wu et al. (2023) introduced adversarial structural biases to attention matrices, though at the cost of increased training complexity.

Wornow et al. (2023) injected Gaussian noise into the latent space of an encoder-decoder model for better image captioning, while Yuan et al. (2022) perturbed hidden representations during fine-tuning to marginally improve language model performance. Most notably, Jain et al. (2023) introduced NEFTune, which adds calibrated uniform noise to embedding vectors during fine-tuning—resulting in significant improvements for models like LLaMA-1 and LLaMA-2. Inspired by these efforts, we compare our method with NEFTune and propose a new approach that directly perturbs the attention scores, encouraging the model to learn richer contextual dependencies across sequences. Here, We investigate augmenting the self-attention scores—central to modeling event dependencies —by injecting and smoothing adaptive Gaussian noise. Unlike prior methods that perturb embeddings or hidden states, our approach directly improves attention behavior without changing the model architecture, enhancing the learned representation in a lightweight and effective way.

## 4 Methods

### 4.1 Smoothed Noise Injection Self-attention Augmentation

In this subsection, we introduce, Smoothed Noise Injection Self-attention Augmentation (SNSA), a simple yet effective two-step augmentation technique designed to improve the learned representations in Transformer models by directly augmenting the attention scores during fine-tuning (Algorithm 1). This method enhances attention dynamics without modifying the computational graph, making it compatible with any pre-trained Transformer encoder.

SNSA operates by first injecting adaptive Gaussian noise into the attention score matrix and then applying a smoothing operation using a Gaussian kernel. This process encourages the model to explore the attention patterns and strengthens context modeling. The augmented attention is computed as:

$$\text{SNSA} = \left( (A_h + \sim \mathcal{N}(\mu, \sigma_{GN}^2)) * n_{\sigma_{eh}} \right) V \tag{5}$$

Here, the Gaussian noise $\mathcal{N}(\mu, \sigma_{\text{GN}}^2)$ is computed adaptivly based on the learned attention during training:

$$\mu = \frac{1}{n^2} \sum_{i=0}^{n-1} \sum_{j=0}^{n-1} A_{i,j} \tag{6}$$

$$\sigma_{\text{GN}} = \sqrt{\frac{1}{n-1} \sum_{i=0}^{n-1} \sum_{j=0}^{n-1} (A_{i,j} - \mu)^2} \tag{7}$$

The smoothing kernel $n_{\sigma_{\text{eh}}}[i, j]$ is a 2D Gaussian distribution:

$$n_{\sigma_{\text{eh}}}[i, j] = \frac{1}{2\pi\sigma_{\text{eh}}^2} e^{-\frac{1}{2}\left(\frac{i^2 + j^2}{\sigma_{\text{eh}}^2}\right)} \tag{8}$$

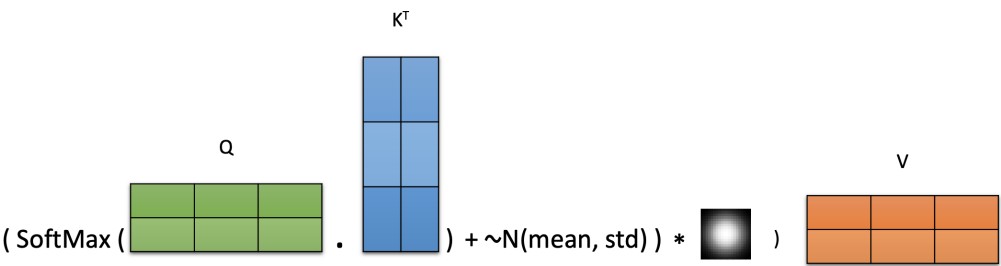

Figure 2: Smoothed Noise injection Self-attention Augmentation (SNSA) mechanism

---

**Algorithm 1** A Transformer Encoder with SNSA

---

**Input**: $D_{\text{fine-tuning}} = \{(X_i, y_i)\}_1^N$ tokenized dataset, embedding layer $\text{emb}(\cdot)$, attention score matrix $A_h$, normal noise $\mathcal{N}(\mu, \sigma^2_{\text{GN}})$, two-dimensional Gaussian noise $n_{\sigma_{\text{eh}}}$, rest of the model $f(\cdot)$
**Parameter**: Normal noise $\mu, \sigma^2_{\text{GN}}$ calculated from $A_h$, event horizon hyperparameter $\sigma_{\text{eh}}$ based on the data charecterstic needs to adjust the smoothing noise

1: Initialize $\theta$ from a pre-trained model
2: **repeat**
3:     Sample $(X_i, y_i) \sim D_{\text{fine-tuning}}$
4:     $X_{\text{emb}} \leftarrow \text{emb}(X_i)$
5:     **for** each Attention Head $A_h$ in Transformer Block **do**
6:         $A_h(X_{\text{attn}}) \leftarrow A_h(X_{\text{emb}}) + \mathcal{N}(\mu, \sigma^2_{\text{GN}})$
7:         $A_h(X_{\text{attn}}) \leftarrow \text{Convolve}(A_h(X_{\text{attn}}), n_{\sigma_{\text{eh}}})$
8:         $H_h(X_{\text{attn}}) \leftarrow A_h(X_{\text{attn}})V$
9:     **end for**
10:    $\text{MultiHead}(H) \leftarrow \text{concat}(H_0(X_{\text{attn}}), \ldots, H_h(X_{\text{attn}}))$
11:    $\hat{y}_i \leftarrow f(\text{MultiHead}(H))$
12:    $\theta \leftarrow \text{opt}(\theta, \text{loss}(\hat{y}_i, y_i))$
13: **until** Stopping criteria met or maximum iterations reached

---

where $\sigma_{\text{eh}}$ is a tunable hyperparameter representing the event horizon, controlling and adjusting the extent of the smoothing. The convolution operation $*$ applies this kernel over the noise-augmented attention matrix:

$$f[i,j] * n_\sigma[i,j] = \frac{1}{2\pi\sigma^2} \sum_{m=1}^{k} \sum_{n=1}^{k} e^{-\frac{1}{2}\left(\frac{m^2+n^2}{\sigma^2}\right)} f[i-m, j-n] \tag{9}$$

where $k = 2\pi\sigma$ is the kernel size.

This smoothing step modulates the added noise, reinforcing stronger attention patterns while allowing for broader exploration in attentions space. The noise parameters $\mu$ and $\sigma_{\text{GN}}$ are computed independently for each attention head to preserve head-specific attention dynamics during training. Figure 2 illustrates the full SNSA mechanism.

Adding adaptive Gaussian noise $\sim \mathcal{N}(\mu, \sigma^2_{GN})$ to the attention scores helps the model escape sub-optimal solutions and promotes learning more diverse interactions between events. The subsequent Gaussian convolution adjusts the magnitude and distribution of the injected noise, encouraging the model to focus on more meaningful and effective attention patterns. A detailed mathematical justification of how this augmentation promotes richer and more flexible attention modeling is provided in Appendix A.5.

During inference, stochasticity from the added noise is removed by replacing it with its expected value $\mu$, ensuring deterministic predictions:

$$\text{SNSA} = ((A_h + \mu) * n_{\sigma_{eh}})V \tag{10}$$

The computational complexity of SNSA is $O(n^2)$ (for more details, see the technical appendix A.9), and since it's primarily used during fine-tuning with limited labeled samples, the additional cost is negligible.

## 5 Experiments

### 5.1 Datasets

In our study, we utilized medical data from two sources: the MIMIC-IV Johnson et al. (2020) hosp module and the Malmö Diet and Cancer Cohort (MDC) Berglund et al. (1993) dataset, approved by the Ethics Review Board of Sweden (Dnr 2023-00503-01). Each EHR trajectory represents a sequence of temporally structured health events. The MIMIC-IV dataset includes 173,000 patient records across 407,000 visits from 2008 to 2019, with 10.6 million medical codes. The MDC dataset, from a cohort study in Sweden, comprises 30,000 individuals with 531,000 visits from 1992 to 2020, offering a more extended patient history—257 codes per patient on average, compared to MIMIC-IV's 61. To ensure consistency, we used only ICD and ATC codes, the only types available in MDC at the beginning, aligning with prior work like Med-BERT on diagnosis codes for risk prediction.

Both datasets use ICD and ATC codes for disease and medication classification. We randomly split each cohort into 70% for pre-training, 20% for fine-tuning, and 10% for testing. After preprocessing, MIMIC-IV had 2,195 unique ICD-9 and 137 ATC-5 codes, while MDC had 1,558 ICD-10 and 111 ATC-5 codes. To assess the generalizability and robustness of our results, the fine-tuning dataset was split into 5 folds. The model was fine-tuned on 4 folds with early stopping on the remaining fold, repeated 5 times with different validation sets. We reported the mean and standard deviation of the AUC on the unseen test dataset. For details, refer to the dataset availability, specifications and implementation details in the technical appendix A.4,A.2.

### 5.2 Problem Formulation

Each dataset $D$ comprises a set of patients $P$, $D = \{P^1, P^2, \ldots, P^{|D|}\}$. In our study, we considered a total of $|D| = 172,980$ patients for MIMIC-IV and $|D| = 29,664$ patients for the MDC cohort. We represent each patient's longitudinal medical trajectory through a structured set of visit encounters as a sequence of events. This representation is denoted as $P^i = \{V_1^i, V_2^i, \ldots, V_O^i\}$, where $O$ represents the total number of visit encounters for patient $i$. Each visit $V_j^i = I_j \cup M_j$ is the union of all diagnosis codes $I_j \subset I$ and prescribed medications $M_j \subset M$ that are recorded for the $P^i$ at visit $V_j^i$. To reduce sparsity, we excluded less frequently occurring medical codes and retained only the initial 4 digits of ICD and ATC codes.

To guide the model in understanding changes in encounter times and the structure of each patient's trajectory, similar to BERT, we employed special tokens. A $[CLS]$ token is placed at the beginning of each patient's trajectory, while a $[SEP]$ token is inserted between visits. Each visit represents a set of diagnoses and medications recorded within a specific time span, and the $[SEP]$ token separates the sets of medical codes from one visit to the next. Consequently, each patient's trajectory is represented as $P^i = \{[CLS], V_1^i, [SEP], V_2^i, [SEP], \ldots, V_O^i, [SEP]\}$, providing the model with valuable context for analysis and prediction.

Here, we evaluated our models on 3 downstream tasks $e_{dt}$ (Heart Failure (HF), Alzheimer Disease (AD), Prolonged Length of Stay on the next visit (PLS) predictions), where the model predicts the incidence of the first HF ($I_{N=HF}$) or AD ($I_{N=AD}$) ICD codes or the presence of PLS ($PLS_N = 1$) on the $N^{th}$ visit, given the patient's previous history of medical codes, $[V_1^i : V_{N-1}^i]$, as a sequence of temporally structured health events:

$$\mathbb{P}(e_{dt} \in V^N \mid P^i = \{[CLS], V_1^i, [SEP], V_2^i, [SEP], \ldots, V_{N-1}^i, [SEP]\}) \tag{11}$$

For each patient's trajectory, if there were no occurrences of the target events $e_{dt}$, it is considered a negative case; otherwise, we exclude the first visit with the target and all subsequent visits and consider it a positive case. All ATC codes related to HF treatment are excluded to avoid timing-related noise and non-trivial predictions. Initially, models exhibited bias toward longer visit histories, confounding risk predictions. To

address this, we excluded trajectories with fewer than 30 visits in the MDC dataset and fewer than 10 visits in the MIMIC-IV dataset. This ensured balanced visit histories between positive and negative cases, resulting in averages of 19 visits in the MDC dataset and 9 visits in the MIMIC-IV dataset, aligning with their overall dataset averages prior to preprocessing. Table 1 summarizes the number of positive and negative cases after these preprocessing steps.

Table 1: Number of positive and negative labeled samples in each downstream task.

| Task | #Positive labels | #Negative labels |
|---|---|---|
| PLS prediction | 2,429 | 6,360 |
| HF prediction on the MIMIC IV | 243 | 641 |
| AD prediction | 245 | 2,628 |
| HF prediction on the MDC | 103 | 301 |

### 5.3 List of Models

To thoroughly investigate the impact of the proposed SNSA augmentation, we compared the performance of following conventional and deep learning models on downstream tasks of HF, AD, and PLS prediction using both the MDC and MIMIC-IV datasets. These models were trained either from scratch or initiated from pre-trained weights, fine-tuned on the fine-tuning dataset, and evaluated on the test dataset. We set the tunable event horizon parameter to $\sigma_{eh} = 1.0$ (kernel size = 6) for the SNSA on the MDC dataset and $\sigma_{eh} = 0.33$ (kernel size = 2) on the MIMIC IV after fine-tuning on the fine-tuning dataset. Except fir HF prediction in the MDC, different $\sigma_{eh}$, slightly changes the SNSA performance. For more details see technical appendix A.3.

**Models with Proposed NSA/SNSA**

- **Transformer with SNSA**: This model incorporates SNSA into all self-attention heads of a randomly initialized Transformer.

- **Transformer pre-trained on MLM with Noise Injection Self-attention Augmentation (NSA)**: In this approach, $\mathcal{N}(\mu, \sigma_{GN}^2)$ (normal noise with adaptive parameters) is added to all self-attention heads of a pre-trained Transformer. This experiment allows us to isolate the impact of the noise injection from the smoothing effect of Gaussian convolution.

- **Transformer pre-trained on MLM with SNSA**: This model incorporates SNSA into all self-attention heads of the pre-trained Transformer.

Baseline model details are provided in Appendix A.11.

### 5.4 Evaluation on downstream tasks

The results are summarized in Table 2 and suggest that adding SNSA improves the AUC of pre-trained Transformers, potentially positioning them as one of the state-of-the-art methods for outcome prediction on temporal structured health data. Specifically, on the MDC dataset, the AUC for HF and AD prediction increased to 74.5% and 73.2%, respectively, while on the MIMIC-IV dataset, the AUC for HF prediction reached 87.2%. The addition of SNSA resulted in statistically significant improvements for HF prediction on both the MDC and MIMIC-IV datasets for the MLM pre-trained Transformer. Furthermore, the improvement in AD prediction was considerable, showcasing the effectiveness of SNSA augmentation. However, incorporating SNSA did not significantly alter the performance of PLS prediction. Additionally, applying GLA to randomly initialized Transformers boosted the AUC for PLS prediction to 60.2%, with negligible effects on other downstream tasks. To delve deeper into the impact of each noise injection and smoothing augmentation term, we solely added the normal noise to the pre-trained Transformer. This experiment revealed that the noise injection alone had a more pronounced effect on downstream tasks in the MIMIC

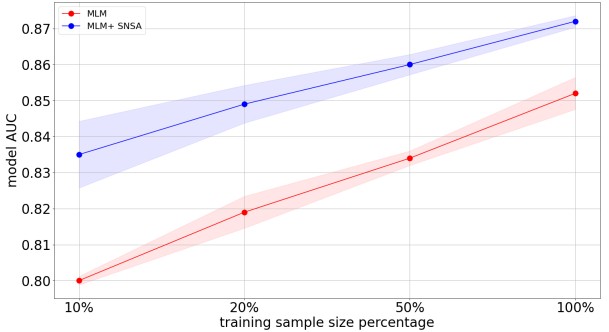
(a) AUC values for HF prediction across various fine-tuning sample sizes on the test dataset in MIMIC IV.

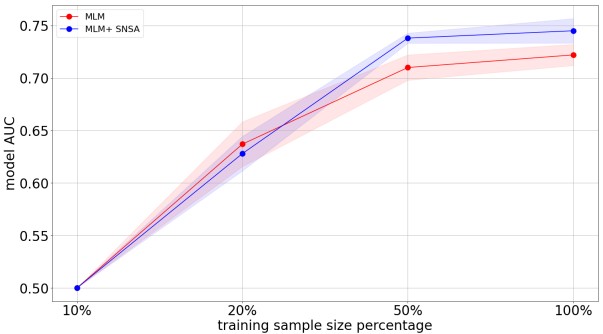
(b) AUC values for HF prediction across various fine-tuning sample sizes on the test dataset in the MDC.

Figure 3: SNSA's impact on HF prediction across fine-tuning sample sizes in MIMIC IV and MDC datasets.

dataset, whereas the combined (SNSA) terms exhibited greater impacts on the downstream tasks in the MDC dataset, particularly associated with its longer sequences.

Table 2: Average AUC values (%) and standard deviation for different methods for the HF prediction, AD prediction, and PLS prediction downstream tasks on the test datasets.

| Model / Downstream Task | HF prediction (MDC) | AD prediction (MDC) | HF prediction (MIMIC-IV) | PLS prediction (MIMIC-IV) |
|---|---|---|---|---|
| Transformer | 71.4 (0.5) | 70.5 (0.8) | 84.2 (1.4) | 54.4 (0.8) |
| Transformer+ SNSA | 72.1 (2.7) | 70.4 (0.6) | 83.2 (2.5) | 60.2 (1.2) |
| Transformer pre-trained on MLM | 72.2 (2.5) | 72.2 (1.1) | 85.2 (1.1) | 60.3 (1.3) |
| Transformer pre-trained on MLM+ NSA | 72.6 (1.9) | 71.4 (1.0) | 86.5 (1.2) | **60.7 (0.6)** |
| Transformer pre-trained on MLM+ SNSA | **74.5 (2.9)** | **73.2 (0.3)** | **87.2 (0.4)** | 60.3 (0.7) |

## 5.5 Performance boost on data insufficiency

One of the advantages of using pre-trained Transformers is their robustness and performance in situations of data insufficiency, observed in both NLP (Brown et al., 2020) and temporal health data (Rasmy et al., 2021). Here, we investigated the effect of applying SNSA on model performance for HF prediction with reduced data sample sizes. We decreased the fine-tuning sample size to 50%, 20%, and 10%, respectively. The performance of the pre-trained Transformer with and without SNSA, was compared on both the MDC and MIMIC-IV datasets. Figure 3a shows that SNSA improves the model performance by around 3% in HF prediction on the MIMIC-IV dataset across all data sample sizes. Similarly, Figure 3b demonstrates that SNSA consistently outperforms the baseline in HF prediction on the MDC dataset, even with a 50% reduction in training samples. However, its superiority diminishes with less data.

## 5.6 VS Hidden Representation Augmentation

We first compared SNSA with other hidden representation augmentation methods proposed for augmenting different layers of pre-trained Transformers. Specifically, we assess the impact of injecting noise into various components of the network, such as hidden layers and feedforward modules, as explored in works like HyPe (Yuan et al., 2022) and Neftune (Jain et al., 2023). Our objective is to evaluate whether augmenting self-attention scores, where contextual dependencies are explicitly encoded, is more effective than augmenting other internal representations.

As shown in Table 3, although NefTune Jain et al. (2023) enhances the performance of pre-trained Transformers in HF prediction across both datasets, SNSA consistently outperforms both NefTune and feedforward noise augmentation in predicting outcomes. While SNSA demonstrates superior performance in this context, NefTune has the advantage of being computationally lighter. However, since both methods are applied during fine-tuning, the computational demands are not a significant concern.

### 5.7 VS Naive masking

Randomly masking the attention score matrix during training can be seen as an extreme form of NSA augmentation. Instead of adding normal noise to perturb relationships between events in a sequence, naive masking directly disrupts these relationships by summing each element with 0 or $-A_{h_{i,j}}$, effectively breaking the connections between tokens. We compared our method with naive self-attention masking, as described by Wu et al. (2023), which introduces a bias in the structure of self-attentions:

$$A_h = \text{softmax}\left(\frac{Q_h K_h^T}{\sqrt{d_k}} + M\right), \quad M \in \{0, -\infty\}^{N \times N},\tag{12}$$

where $M_{i,j} = -\infty$ with $p = 0.2$, optimized based on performance on the fine-tuning dataset. We extended it to DropAttention (Zehui et al., 2019), which expands the mask with a span length $\omega$ and we set $\omega = $ Kernel size. However, neither naive masking nor DropAttention improved the performance of the pre-trained Transformer for HF prediction on the MDC and MIMIC-IV datasets. Instead, these methods only increased the number of training iterations required for convergence (see Table 3). While these techniques can help mitigate overfitting, their overly aggressive regularization often disrupts critical dependencies within sequences, leading to unstable training and poorer overall performance, especially on complex healthcare prediction tasks. In contrast, SNSA introduces controlled perturbations that balance the attention distribution and prevent over-reliance on specific patterns, thereby preserving essential relationships in the data and promoting more robust and effective representations (see Appendix A.7 for a justification of SNSA as a structured variant of dropout).

Table 3: Comparing SNSA with naive masking and other hidden representation augmentation methods. The table shows the average AUC values (%) and standard deviation across HF prediction tasks on the MDC and MIMIC-IV datasets.

| Model / Downstream Task | HF Prediction (MDC) | HF Prediction (MIMIC-IV) |
|---|---|---|
| Transformer pre-trained on MLM | 72.2 (2.5) | 85.2 (1.1) |
| Transformer pre-trained on MLM+ Naive masking | 70.00 (1.5) | 85.1 (0.7) |
| Transformer pre-trained on MLM+ DropAttention | 69.7 (1.1) | 84.9 (1.3) |
| Transformer pre-trained on MLM+ NEFTune($\alpha = 5$) | 73.6 (3.2) | 85.2 (0.7) |
| Transformer pre-trained on MLM+ NEFTune($\alpha = 10$) | 73.1 (1.7) | 85.5 (0.4) |
| Transformer pre-trained on MLM+ noise in the feedforward($\alpha = 5$) | 73.7 (2.2) | 85.0 (1.2) |
| Transformer pre-trained on MLM+ noise in the feedforward($\alpha = 10$) | 72.5 (4.4) | 84.5 (0.8) |
| Transformer pre-trained on MLM+ SNSA | **74.5 (2.9)** | **87.2 (0.4)** |

### 5.8 Effect of SNSA on self-attention behavior

Analyzing self-attention weights and attention score matrices can highlight how Transformers prioritize relationships between events, shedding light on their internal logic and behavior (Clark et al., 2019; Kovaleva et al., 2019; Hao et al., 2021). To assess the effect of SNSA and compare it with normal noise injection (NSA), we analyzed attention score distributions in models fine-tuned on all downstream tasks.

We plotted histograms of attention scores across all heads and samples from the test split, scaling each head's scores to the [0, 1] range (Figure 4). In the bottom row of the figure, we observe that attention scores from the fine-tuned vanilla Transformer tend to cluster near 0 or 1, forming a near-binary (binomial-like) distribution. This pattern suggests overconfidence and limited exploration of dependencies across tokens.

In contrast, the middle row shows that NSA—injecting Gaussian noise during training—broadens the distribution, encouraging attention heads to explore more diverse and weaker connections. This leads to over-lapping attention patterns and increased representation diversity. A mathematical explanation for this phenomenon is provided in Appendix A.5.

The top row demonstrates the effect of SNSA, which combines noise injection with Gaussian smoothing. This operation retains the diversity introduced by noise while stabilizing the attention pattern, restoring smoother and more informative distributions. The smoothing step dampens extreme noise while allowing the model to refine its exploration of differnt interactions.

To further investigate, we visualized the attention score matrices from models fine-tuned on a representative test sample from the HF prediction task on the MDC dataset (Figure 5). Comparing the original and smoothed attention scores, we observe that SNSA promotes broader attention coverage, with activation scores scaled to the [0, 1] range. Figure 5 illustrates an attention head from the first layer, confirming that SNSA leads to more distributed attention patterns. Additional examples from the MIMIC-IV dataset are provided in the appendix A.10.

### 5.8.1    Effect of SNSA on the Receptive Field

The self-attention mechanism is designed to capture both long and short-range dependencies effectively. To quantitatively assess the impact of NSA and SNSA on the receptive field, we plot the median values of attention score matrix $A_h$ for each event with respect to all previous and subsequent events $(i - j, A_{h_{i,j}})$ -$i, j$ are positions of $e_i, e_j$ in the sequence of events- across all test samples for HF and AD predictions on the MDC (Figures 6). Transformers pre-trained on MLM typically allocate more attention weight to recent events, often in a monotonous fashion. Incorporating NSA regularization reduces the steepness of this attention distribution, allowing events to receive more balanced attention, not solely based on their proximity to recent events. Ultimately, applying SNSA, preserves the benefits of NSA by providing a more equal distribution of attention within a local neighborhood, while simultaneously reducing the emphasis on very distant past events. However, it is important to note that raw self-attention values do not fully reveal Transformer behavior, as they are not directly interpretable and require further processing for accurate attribution (Hao et al., 2021; Jain & Wallace, 2019; Serrano & Smith, 2019).

## 6    Conclusion

In this work, we introduced Smoothed Noise Injection Self-Attention Augmentation (SNSA), a simple yet effective method for enhancing the fine-tuning of pre-trained Transformers on temporally structured health-care data. SNSA directly augments the self-attention scores with adaptive Gaussian noise and applies a smoothing convolution using a Gaussian kernel, encouraging the model to explore more diverse attention patterns while preserving critical dependencies.

We demonstrated that pre-trained Transformers, when fine-tuned on limited EHR datasets, often converge to overly sharp attention distributions—overfitting to local patterns and failing to capture broader contextual relationships. Our approach addresses this limitation by injecting controlled stochasticity and smoothing, leading to improved generalization and robustness. Extensive experiments on multiple clinical prediction tasks showed that SNSA consistently outperforms conventional regularization and augmentation techniques.

SNSA offers a plug-and-play augmentation mechanism that operates entirely within the attention computation, requiring no modification to the model architecture or computational graph. This makes it particularly suitable for integration with existing pre-trained models.

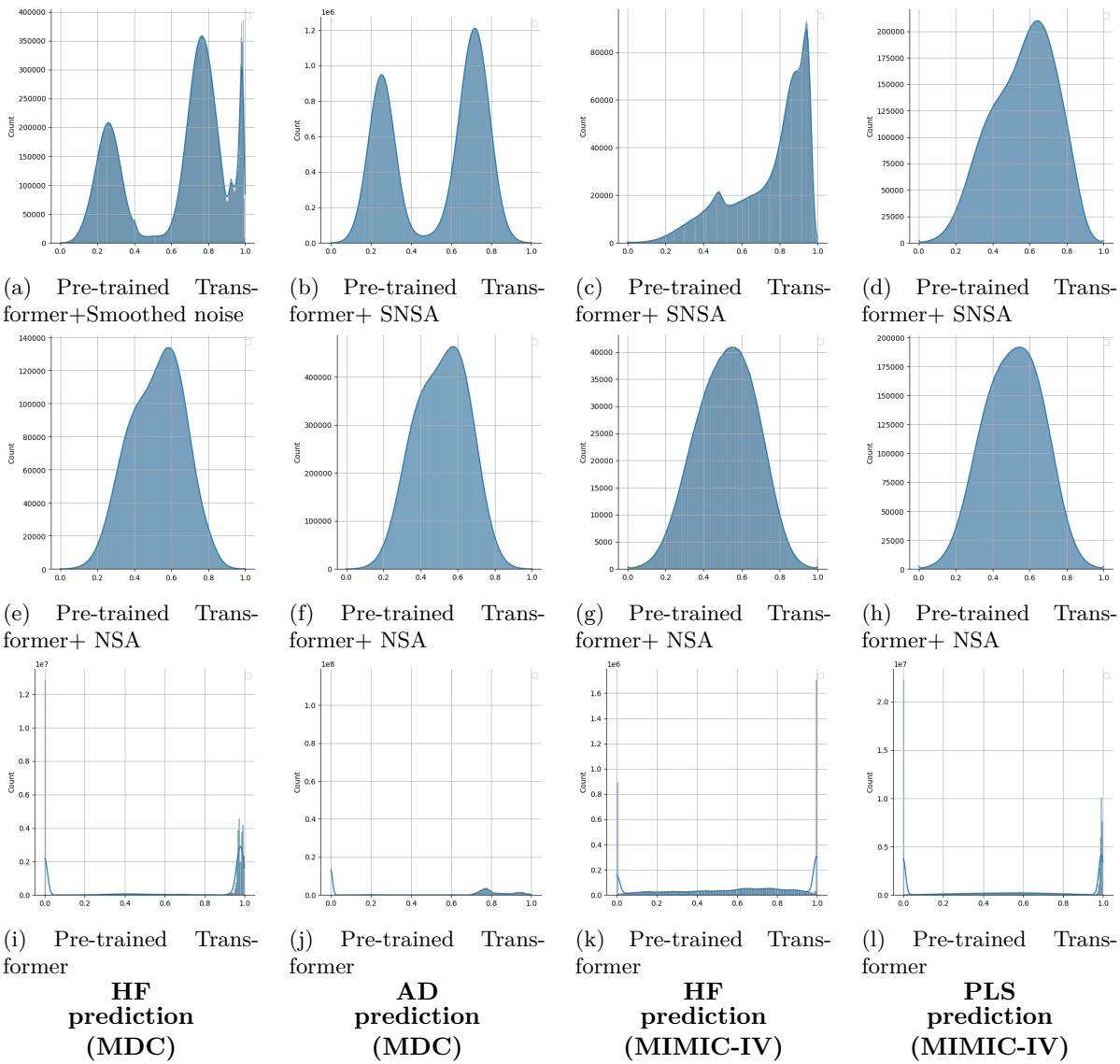

Figure 4: Comparison of the impact of SNSA on self-attention score distributions in fine-tuned models. Attention scores from each head are individually scaled to the [0, 1] range before plotting their distributions.

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

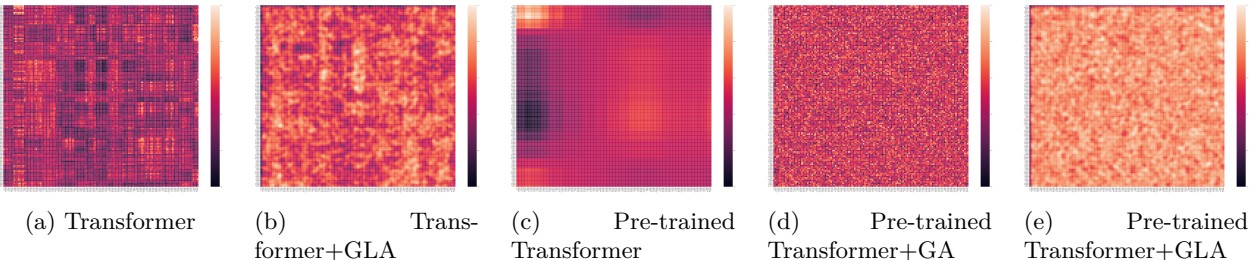

(a) Transformer (b) Transformer+GLA (c) Pre-trained Transformer (d) Pre-trained Transformer+GA (e) Pre-trained Transformer+GLA

Figure 5: Comparing the impact of GLA on the self-attention score weights for five fine-tuned models on HF prediction on the MDC dataset for a specific test sample. Here, the attention scores are scaled within 0 and 1.

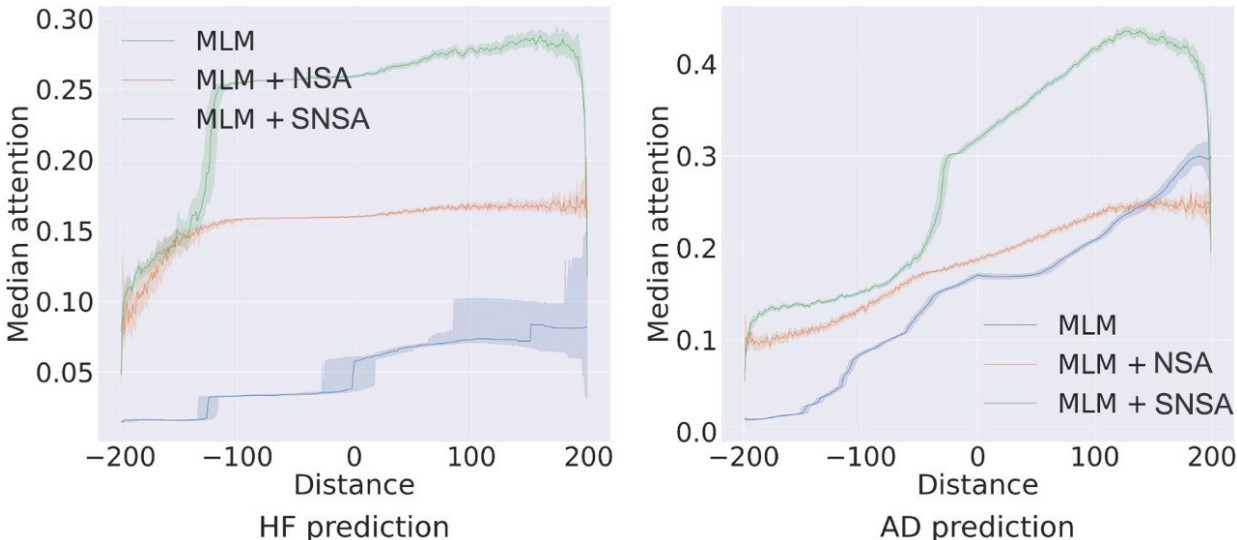

Figure 6: Impact of SNSA on the receptive field of the self-attentions for HF and AD prediction on the MDC dataset.

Heloísa Oss Boll, Ali Amirahmadi, Mirfarid Musavian Ghazani, Wagner Ourique de Morais, Edison Pignaton de Freitas, Amira Soliman, Kobra Etminani, Stefan Byttner, and Mariana Recamonde-Mendoza. Graph neural networks for clinical risk prediction based on electronic health records: A survey. *Journal of Biomedical Informatics*, pp. 104616, 2024.

Rishi Bommasani, Drew A Hudson, Ehsan Adeli, Russ Altman, Simran Arora, Sydney von Arx, Michael S Bernstein, Jeannette Bohg, Antoine Bosselut, Emma Brunskill, et al. On the opportunities and risks of foundation models. *arXiv preprint arXiv:2108.07258*, 2021.

Tom Brown, Benjamin Mann, Nick Ryder, Melanie Subbiah, Jared D Kaplan, Prafulla Dhariwal, Arvind Neelakantan, Pranav Shyam, Girish Sastry, Amanda Askell, et al. Language models are few-shot learners. *Advances in neural information processing systems*, 33:1877–1901, 2020.

Alexander Camuto, Matthew Willetts, Umut Simsekli, Stephen J Roberts, and Chris C Holmes. Explicit regularisation in gaussian noise injections. *Advances in Neural Information Processing Systems*, 33:16603–16614, 2020.

Jiaao Chen, Zichao Yang, and Diyi Yang. Mixtext: Linguistically-informed interpolation of hidden space for semi-supervised text classification. *arXiv preprint arXiv:2004.12239*, 2020.

Xinlei Chen, Saining Xie, and Kaiming He. An empirical study of training self-supervised vision transformers. In *Proceedings of the IEEE/CVF international conference on computer vision*, pp. 9640–9649, 2021.

Edward Choi, Zhen Xu, Yujia Li, Michael Dusenberry, Gerardo Flores, Emily Xue, and Andrew Dai. Learning the graphical structure of electronic health records with graph convolutional transformer. In *Proceedings of the AAAI conference on artificial intelligence*, volume 34, pp. 606–613, 2020.

Kevin Clark, Urvashi Khandelwal, Omer Levy, and Christopher D Manning. What does bert look at? an analysis of bert's attention. *arXiv preprint arXiv:1906.04341*, 2019.

Jacob Devlin, Ming-Wei Chang, Kenton Lee, and Kristina Toutanova. Bert: Pre-training of deep bidirectional transformers for language understanding. *arXiv preprint arXiv:1810.04805*, 2018.

Jiayu Ding, Shuming Ma, Li Dong, Xingxing Zhang, Shaohan Huang, Wenhui Wang, Nanning Zheng, and Furu Wei. Longnet: Scaling transformers to 1,000,000,000 tokens. *arXiv preprint arXiv:2307.02486*, 2023.

Alexey Dosovitskiy, Lucas Beyer, Alexander Kolesnikov, Dirk Weissenborn, Xiaohua Zhai, Thomas Unterthiner, Mostafa Dehghani, Matthias Minderer, Georg Heigold, Sylvain Gelly, et al. An image is worth 16x16 words: Transformers for image recognition at scale. *arXiv preprint arXiv:2010.11929*, 2020.

Lin Lawrence Guo, Ethan Steinberg, Scott Lanyon Fleming, Jose Posada, Joshua Lemmon, Stephen R Pfohl, Nigam Shah, Jason Fries, and Lillian Sung. Ehr foundation models improve robustness in the presence of temporal distribution shift. *Scientific Reports*, 13(1):3767, 2023.

Yaru Hao, Li Dong, Furu Wei, and Ke Xu. Self-attention attribution: Interpreting information interactions inside transformer. In *Proceedings of the AAAI Conference on Artificial Intelligence*, volume 35, pp. 12963–12971, 2021.

Ali Hassani, Steven Walton, Nikhil Shah, Abulikemu Abuduweili, Jiachen Li, and Humphrey Shi. Escaping the big data paradigm with compact transformers. *arXiv preprint arXiv:2104.05704*, 2021.

Ali Hassani, Steven Walton, Jiachen Li, Shen Li, and Humphrey Shi. Neighborhood attention transformer. In *Proceedings of the IEEE/CVF conference on computer vision and pattern recognition*, pp. 6185–6194, 2023.

Neel Jain, Ping-yeh Chiang, Yuxin Wen, John Kirchenbauer, Hong-Min Chu, Gowthami Somepalli, Brian R Bartoldson, Bhavya Kailkhura, Avi Schwarzschild, Aniruddha Saha, et al. Neftune: Noisy embeddings improve instruction finetuning. *arXiv preprint arXiv:2310.05914*, 2023.

Sarthak Jain and Byron C Wallace. Attention is not explanation. *arXiv preprint arXiv:1902.10186*, 2019.

Alistair Johnson, Lucas Bulgarelli, Tom Pollard, Steven Horng, Leo Anthony Celi, and Roger Mark. Mimic-iv. *PhysioNet. Available online at: https://physionet. org/content/mimiciv/1.0/(accessed August 23, 2021)*, 2020.

Kyung Geun Kim and Byeong Tak Lee. Self-attention with temporal prior: can we learn more from the arrow of time? *Frontiers in Artificial Intelligence*, 7:1397298, 2024.

Kezhi Kong, Guohao Li, Mucong Ding, Zuxuan Wu, Chen Zhu, Bernard Ghanem, Gavin Taylor, and Tom Goldstein. Robust optimization as data augmentation for large-scale graphs. In *Proceedings of the IEEE/CVF Conference on Computer Vision and Pattern Recognition*, pp. 60–69, 2022.

Olga Kovaleva, Alexey Romanov, Anna Rogers, and Anna Rumshisky. Revealing the dark secrets of bert. *arXiv preprint arXiv:1908.08593*, 2019.

Zhenzhong Lan, Mingda Chen, Sebastian Goodman, Kevin Gimpel, Piyush Sharma, and Radu Soricut. Albert: A lite bert for self-supervised learning of language representations. *arXiv preprint arXiv:1909.11942*, 2019.

Mike Lewis, Yinhan Liu, Naman Goyal, Marjan Ghazvininejad, Abdelrahman Mohamed, Omer Levy, Ves Stoyanov, and Luke Zettlemoyer. Bart: Denoising sequence-to-sequence pre-training for natural language generation, translation, and comprehension. *arXiv preprint arXiv:1910.13461*, 2019.

Yikuan Li, Shishir Rao, José Roberto Ayala Solares, Abdelaali Hassaine, Rema Ramakrishnan, Dexter Canoy, Yajie Zhu, Kazem Rahimi, and Gholamreza Salimi-Khorshidi. Behrt: transformer for electronic health records. *Scientific reports*, 10(1):7155, 2020.

Pengfei Liu, Weizhe Yuan, Jinlan Fu, Zhengbao Jiang, Hiroaki Hayashi, and Graham Neubig. Pre-train, prompt, and predict: A systematic survey of prompting methods in natural language processing. *ACM Computing Surveys*, 55(9):1–35, 2023.

Yinhan Liu, Myle Ott, Naman Goyal, Jingfei Du, Mandar Joshi, Danqi Chen, Omer Levy, Mike Lewis, Luke Zettlemoyer, and Veselin Stoyanov. Roberta: A robustly optimized bert pretraining approach. *arXiv preprint arXiv:1907.11692*, 2019.

Yiwen Meng, William Speier, Michael K Ong, and Corey W Arnold. Bidirectional representation learning from transformers using multimodal electronic health record data to predict depression. *IEEE journal of biomedical and health informatics*, 25(8):3121–3129, 2021.

Chao Pang, Xinzhuo Jiang, Krishna S Kalluri, Matthew Spotnitz, RuiJun Chen, Adler Perotte, and Karthik Natarajan. Cehr-bert: Incorporating temporal information from structured ehr data to improve prediction tasks. In *Machine Learning for Health*, pp. 239–260. PMLR, 2021.

Raphael Poulain, Mehak Gupta, and Rahmatollah Beheshti. Few-shot learning with semi-supervised transformers for electronic health records. In *Machine Learning for Healthcare Conference*, pp. 853–873. PMLR, 2022.

Ofir Press, Noah A Smith, and Mike Lewis. Train short, test long: Attention with linear biases enables input length extrapolation. *arXiv preprint arXiv:2108.12409*, 2021.

Alec Radford, Jeffrey Wu, Rewon Child, David Luan, Dario Amodei, Ilya Sutskever, et al. Language models are unsupervised multitask learners. *OpenAI blog*, 1(8):9, 2019.

Laila Rasmy, Yang Xiang, Ziqian Xie, Cui Tao, and Degui Zhi. Med-bert: pretrained contextualized embeddings on large-scale structured electronic health records for disease prediction. *NPJ digital medicine*, 4(1):86, 2021.

Houxing Ren, Jingyuan Wang, Wayne Xin Zhao, and Ning Wu. Rapt: Pre-training of time-aware transformer for learning robust healthcare representation. In *Proceedings of the 27th ACM SIGKDD conference on knowledge discovery & data mining*, pp. 3503–3511, 2021.

Sofia Serrano and Noah A Smith. Is attention interpretable? *arXiv preprint arXiv:1906.03731*, 2019.

Jianlin Su, Murtadha Ahmed, Yu Lu, Shengfeng Pan, Wen Bo, and Yunfeng Liu. Roformer: Enhanced transformer with rotary position embedding. *Neurocomputing*, 568:127063, 2024.

Hugo Touvron, Matthieu Cord, Matthijs Douze, Francisco Massa, Alexandre Sablayrolles, and Hervé Jégou. Training data-efficient image transformers & distillation through attention. In *International conference on machine learning*, pp. 10347–10357. PMLR, 2021.

Ashish Vaswani, Noam Shazeer, Niki Parmar, Jakob Uszkoreit, Llion Jones, Aidan N Gomez, Łukasz Kaiser, and Illia Polosukhin. Attention is all you need. *Advances in neural information processing systems*, 30, 2017.

Michael Wornow, Yizhe Xu, Rahul Thapa, Birju Patel, Ethan Steinberg, Scott Fleming, Michael A Pfeffer, Jason Fries, and Nigam H Shah. The shaky foundations of large language models and foundation models for electronic health records. *npj Digital Medicine*, 6(1):135, 2023.

Hongqiu Wu, Ruixue Ding, Hai Zhao, Pengjun Xie, Fei Huang, and Min Zhang. Adversarial self-attention for language understanding. In *Proceedings of the AAAI Conference on Artificial Intelligence*, volume 37, pp. 13727–13735, 2023.

Cao Xiao, Edward Choi, and Jimeng Sun. Opportunities and challenges in developing deep learning models using electronic health records data: a systematic review. *Journal of the American Medical Informatics Association*, 25(10):1419–1428, 2018.

Hongyi Yuan, Zheng Yuan, Chuanqi Tan, Fei Huang, and Songfang Huang. Hype: Better pre-trained language model fine-tuning with hidden representation perturbation. *arXiv preprint arXiv:2212.08853*, 2022.

Lin Zehui, Pengfei Liu, Luyao Huang, Junkun Chen, Xipeng Qiu, and Xuanjing Huang. Dropattention: A regularization method for fully-connected self-attention networks. *arXiv preprint arXiv:1907.11065*, 2019.

Chen Zhu, Yu Cheng, Zhe Gan, Siqi Sun, Tom Goldstein, and Jingjing Liu. Freelb: Enhanced adversarial training for natural language understanding. *arXiv preprint arXiv:1909.11764*, 2019.

Weicheng Zhu and Narges Razavian. Variationally regularized graph-based representation learning for electronic health records. In *Proceedings of the Conference on Health, Inference, and Learning*, pp. 1–13, 2021.

# A    Appendix

## A.1    Implementation Details

The code compatible with the public MIMIC-IV dataset is provided in the Code Appendix.

The Transformer encoder was tuned on the MLM pretraining task based on the HF prediction results on the validation dataset. We experimented with different hyperparameter configurations, including the number of attention heads set to 1, 3, 5, and 8, hidden dimensions of 16, 36, and 64, and Transformer encoder layers set to 1, 4, and 8. The best-performing configuration consisted of five attention heads, a hidden dimension of 36, and a single Transformer encoder layer.

For MLM pretraining, we followed the approach in Poulain et al. (2022), where 15% of medical tokens were randomly selected for modification. Of these masked tokens, 80% were replaced with the `[MASK]` token, 10% were replaced with a randomly selected different medical token, and the remaining 10% were left unchanged.

### A.1.1    Pretraining on MLM

For pretraining, we used the pretraining splits for each cohort as described in Section A.2. The input sequence length was determined as the 0.7 percentile of the distribution of sequence lengths in the pretraining dataset, which resulted in 131 tokens for the MDC dataset and 65 tokens for the MIMIC-IV dataset. A sliding window approach with a stride of one was used to augment data during the MLM pretraining phase. This resulted in approximately 313,000 training samples and 33,000 validation samples for MDC, while MIMIC-IV contained around 1.245 million training samples and 140,000 validation samples. The vocabulary sizes were 1,675 for MDC and 2,338 for MIMIC-IV. The models were pre-trained for 50 epochs, after which the training and validation losses plateaued, as shown in Figure 7.

### A.1.2    Training and Fine-Tuning on Downstream Task Prediction

For fine-tuning, token representations were aggregated using a GRU layer before being fed into a classifier. The optimizer and the learning rate and coefficient, warm up steps and dropout were optimized from [SGD, Adam, AdamW], [1e-5,2e-5,5e-5,7e-5], co=[1,.95,.9,.8] and warmup=[0,.05,.1,.2]*total steps and dropout=[0, .1, .2,.4].The model was trained using the Adam (for 1 layer and ADAMW for 4 and 8 layer) optimizer with a layer-wise learning rate decay coefficient of 0.9 and an initial learning rate of $6 \times 10^{-5}$. The input sequence length was set to 200 medical codes.

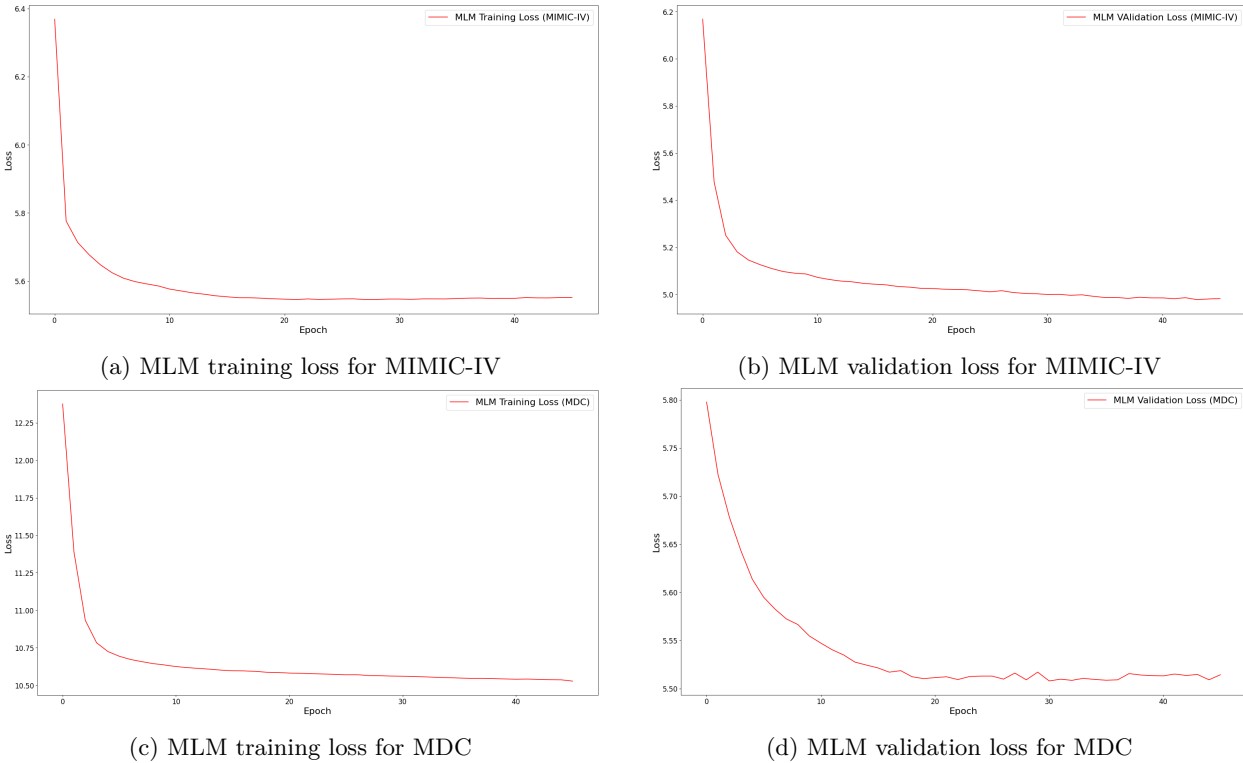

(a) MLM training loss for MIMIC-IV

(b) MLM validation loss for MIMIC-IV

(c) MLM training loss for MDC

(d) MLM validation loss for MDC

Figure 7: MLM training and validation loss for MIMIC-IV and MDC datasets during training.

### A.1.3 Cross-Validation

For model evaluation, five-fold cross-validation was conducted. The dataset was divided into five folds, and the model was fine-tuned on four folds while early stopping was applied using the remaining fold. This process was repeated five times with different validation sets, and the reported results include the mean and standard deviation of the AUC on the unseen test dataset.

### A.2 Dataset specifications

We used medical data from two sources: the Medical Information Mart for Intensive Care IV (MIMIC-IV) Johnson et al. (2020) hosp module, and the Malmö Diet and Cancer Cohort (MDC) Berglund et al. (1993) dataset, approved by the Ethics Review Board of Sweden (Dnr 2023-00503-01). Each EHR trajectory represents a sequence of events of temporal structured health data. The MIMIC-IV hosp module is a comprehensive collection of inpatient EHR trajectories, containing approximately 173,000 patient records documented during 407,000 visits spanning from 2008 to 2019. This dataset includes a total of 10.6 million medical codes representing diagnoses and medications.

On the other hand, the MDC dataset originates from a prospective cohort study conducted in Sweden. It consists of around 30,000 individuals residing in Malmö between 1991 and 1996, with records of both inpatient and outpatient visits spanning from 1992 to 2020, resulting in a total of 531,000 visits. While the MDC dataset has fewer overall samples, it provides a more extensive patient history, with an average of 257 codes per patient compared to MIMIC-IV's 61.

Both datasets use the International Statistical Classification of Diseases and Related Health Problems (ICD) and Anatomical Therapeutic Chemical Code (ATC) for disease and medication classification, respectively, in a hierarchical format.

To facilitate our self-supervised pre-training, supervised fine-tuning, and final testing, we partitioned the extracted cohort randomly into three subsets: 70%, 20%, and 10%, respectively. Despite being characterized by extensive sparsity, preprocessing resulted in 2,195 unique ICD-9 and 137 unique ATC-5 codes for the MIMIC-IV dataset and 1,558 unique ICD10 and 111 unique ATC-5 codes for the MDC dataset.

Table 4: MIMIC-IV dataset summary statistics.

|  | Pre-training dataset | Fine-tuning dataset | Test dataset | Total dataset |
|---|---|---|---|---|
| #patients | 121 K | 36 K | 16 K | 173 K |
| #visits | 285 K | 86 K | 37 K | 408 K |
| #Medical codes | 7.451 M | 2.234 M | 937 K | 10.622 M |

Table 5: MDC dataset summary statistics.

|  | Pre-raining dataset | Fine-tuning dataset | Test dataset | Total dataset |
|---|---|---|---|---|
| #patients | 21 K | 6 K | 3 K | 30 K |
| #visits | 373 K | 107 K | 52 K | 531 K |
| #Medical codes | 5.339 M | 1.554 K | 741 K | 7.634 M |

### A.3 Effect of Varying $\sigma_{eh}$

Table 6 shows the AUC scores on the validation split for each downstream task using a range of $\sigma_{eh}$ values and their corresponding Gaussian kernel sizes. Overall, SNSA shows robust performance across different $\sigma_{eh}$ values, with optimal results achieved consistently within each dataset. Notably, for the MDC dataset, performance on the HF prediction task is more sensitive to changes in $\sigma_{eh}$, while other tasks remain relatively stable.

Table 6: Effect of different $\sigma_{eh}$ values (and corresponding kernel sizes) on validation split AUC across four downstream tasks.

| Task / $\sigma_{eh}$ | 0.3 (k=2) | 0.6 (k=4) | 1.0 (k=6) | 1.5 (k=10) | 3.1 (k=20) | 6.5 (k=40) | 13.4 (k=80) | 20.0 (k=120) |
|---|---|---|---|---|---|---|---|---|
| HF prediction (MDC) | 0.791 | 0.790 | **0.841** | 0.822 | 0.771 | 0.792 | 0.811 | 0.802 |
| AD prediction (MDC) | 0.800 | 0.788 | **0.804** | 0.782 | 0.791 | 0.803 | 0.796 | 0.778 |
| HF prediction (MIMIC-IV) | **0.923** | 0.910 | 0.914 | 0.914 | 0.901 | 0.910 | 0.918 | 0.914 |
| PLS prediction (MIMIC-IV) | **0.603** | 0.598 | 0.591 | 0.603 | 0.599 | 0.588 | 0.597 | 0.596 |

### A.4 Data availability

The MIMIC-IV data is available on `https://physionet.org/content/mimiciv/2.2/`. The MDC dataset is available upon application and with permission of the Malmo Population-Based Cohorts Joint Database `https://www.malmo-kohorter.lu.se/malmo-cohorts`

### A.5 Mathematical Justification: Impact of Normal Noise on Self-Attention distribution

#### A.5.1 Effect of Noise on Self-Attention

As shown in the bottom row of Fig. 4, attention scores in pre-trained Transformers often converge to extreme values—either close to 0 or 1—after fine-tuning. This suggests that the model makes sharp, deterministic decisions regarding which tokens to attend to. To introduce stochasticity and encourage more flexible learning of dependencies, we inject Gaussian noise into the attention scores during training. In this section, we analyze how this perturbation affects the distribution of attention scores.

#### A.5.2 Original Distribution of Attention Scores

From empirical observations in Fig. 4, we approximate the attention score distribution as bi-modal—concentrated at 0 and 1. This can be modeled as a mixture of two Dirac delta functions:

$$p(A_h) = \alpha\delta(A_h - 0) + (1 - \alpha)\delta(A_h - 1)$$

where $\alpha$ is the weight for the peak at 0, and $1 - \alpha$ is the weight for the peak at 1.

The mean and variance of this distribution are:

$$\mathbb{E}[A_h] = (1 - \alpha)$$
$$\mathbb{E}[A_h^2] = (1 - \alpha)$$

Thus, the variance of the attention scores is:

$$\text{Var}(A_h) = \mathbb{E}[A_h^2] - (\mathbb{E}[A_h])^2 = (1 - \alpha) - (1 - \alpha)^2 = \alpha(1 - \alpha)$$

This gives the standard deviation:

$$\text{std}(A_h) = \sqrt{\alpha(1 - \alpha)}$$

#### A.5.3 Effect of Gaussian Noise Injection

We inject Gaussian noise into the attention scores during training:

$$A_h' = A_h + \epsilon \quad \text{where} \quad \epsilon \sim \mathcal{N}(\mu, \text{std}(A_h)^2)$$

The noise $\epsilon$ is drawn from a normal distribution with $\mu$ mean and variance $\text{std}(A_h)^2 = \alpha(1 - \alpha)$.

Thus, the perturbed distribution becomes:

$$A_h' = \begin{cases} \mathcal{N}(\mu, \alpha(1 - \alpha)), & \text{if } A_h = 0, \\ \mathcal{N}(1 + \mu, \alpha(1 - \alpha)), & \text{if } A_h = 1. \end{cases}$$

Thus, the original binary peaks at 0 and 1 are smoothed into overlapping Gaussian centered at $\mu$ and $1 + \mu$ respectively.

#### A.5.4 Effect on the Distribution of Attention Scores

This transformation shifts the attention distribution from discrete to continuous, promoting diversity in attended interactions:

- When $A_h = 0$, the perturbed attention score $A_h'$ will follow a normal distribution centered at $\mu$ .

- When $A_h = 1$, the perturbed attention score $A_h'$ will follow a normal distribution centered at $1 + \mu$ .

As shown in the middle row of Fig. 4, this noise increases overlap between previously distinct peaks. When rescaled between 0 and 1, the distribution appears more centralized and continuous. Notably, the variance is maximized when $\alpha = \frac{1}{2}$, which results in the greatest overlap and diversity.

### A.5.5 Conclusion

Injecting Gaussian noise shifts the attention score distribution from deterministic and binary to probabilistic and continuous, encouraging the model to explore alternative dependencies. The overlap of noised distributions allows the model to escape rigid patterns and better capture complex, context-dependent relationships. Crucially, our use of an adaptive standard deviation based on the attention scores allows the model to modulate this behavior based on the context, facilitating better generalization and diversity across heads and layers.

### A.5.6 Regularization and Robustness

Beyond promoting attention diversity, noise injection acts as a regularizer with several benefits:

- It prevents overfitting to strong patterns by injecting uncertainty.

- It encourages attention to low-probability events.

- It increases the variance of attention distributions, enabling the model to attend to underrepresented interactions.

- It promotes exploration and robustness

These effects support improved generalization in data-scarce environments and more expressive modeling.

### A.6 Justification: Impact of Gaussian Smoothing

The smoothing operation is used to adjust the added noise.

### A.6.1 Effect of Gaussian Convolution on Attention Matrix

The convolution operation:

$$A_h'' = (A_h + \mathcal{N}(\mu, \sigma_{GN}^2)) * n_{\sigma_{eh}} \tag{13}$$

performs localized averaging of the noisy attention matrix $A_h + \mathcal{N}(\mu, \sigma_{GN}^2)$, where:

$$A_h''(i, j) = \sum_{m=1}^{k} \sum_{n=1}^{k} A_h'(i - m, j - n) \cdot n_{\sigma_{eh}}[m, n] \tag{14}$$

This operation results in three key effects:

1. Noise Reduction: Independent Gaussian noise can introduce high-frequency fluctuations. Convolving with a Gaussian kernel acts as a low-pass filter, reducing such variance.

2. parsity Reduction: In the original attention, many values may be near-zero, creating sparsity. Smoothing fills in values by interpolating between neighbors, producing more continuous transitions.

3. Robust Contextualization: Original patterns are preserved and extended through smoothing

### A.7   Justification for SNSA: A Comparison with Dropout

SNSA can be justified by drawing parallels with dropout regularization, viewing SNSA as an "adaptive" extension of it. Dropout works by randomly setting some of the activations to zero, effectively disconnecting certain nodes during training. This prevents the model from becoming overly dependent on specific neurons and encourages a more robust and generalized representation.

Mathematically, for each attention head $A_h$, dropout can be seen as applying a mask $M$ (where $M$ is a Bernoulli distribution), resulting in the modified attention head $A'_h = M \cdot A_h$. In contrast, GLA applies a more nuanced adjustment:

$$A_h = A_h + \epsilon \sim \mathcal{N}(\mu, \sigma^2) = \left(1 + \frac{\epsilon}{A_h}\right) A_h = P \cdot A_h \tag{15}$$

Here, $P = \left(1 + \frac{\epsilon}{A_h}\right)$ acts as an adaptive perturbation factor. Instead of completely severing connections between tokens (as in dropout), GLA adjusts the attention weights by either amplifying or diminishing the focus between two events. This approach maintains the relationships within the data while still introducing variability.

The random perturbation from GLA forces the attention mechanism to avoid over-reliance on specific patterns by continually adjusting the attention distribution. Consequently, GLA can be seen as a form of ensemble learning, where each perturbation offers a different perspective on the data. This effectively trains multiple versions of the model in parallel, each slightly varied due to the noise, leading to a more robust and generalized final model.

### A.8   Performance boost on data insufficiency

Table 7 presents the numerical results corresponding to the data insufficiency section.

Table 7: Effect of incorporating SNSA into the pre-trained Transformer on AUC performance value of HF prediction across various fine-tunning sample sizes on the test dataset in MIMICIV and the MDC

| Model-dataset / fine-tuning percentage | 10 % | 20% | 50% | 100% |
|---|---|---|---|---|
| MLM-MDC | 0.5 (0) | 0.637 (0.053) | 0.710 (0.030) | 0.722 (0.025) |
| MLM+SNSA-MDC | 0.5 (0) | 0.628 (0.041) | 0.738 (0.012) | 0.745 (0.029) |
| MLM-MIMICIV | 0.800 (.003) | 0.819 (.011) | 0.834 (.005) | 0.852 (.011) |
| MLM+SNSA-MIMICIV | 0.835 (.023) | 0.849 (.013) | 0.860 (.007) | 0.872 (.004) |

### A.9   Computational Complexity and Scalability

SNSA involves two primary operations per attention head: adding iid normal noise to an $n \times n$ attention score matrix and convolving the result with a Gaussian filter. Therefore, the computational complexity of SNSA can be expressed as:

$$O(\text{SNSA}) = O(\text{addition of } n \times n \text{ matrix}) + O(\text{convolution})$$
$$O(\text{SNSA}) = O(n^2) + O(k^2 \times n^2) = O(n^2) + O(n^2) = O(n^2)$$

Here, $n$ represents the input length (where $n = 200$ in our case), and $k$ is the kernel size ($k \ll n$). Since the noise addition and convolution operations are only performed during the fine-tuning phase—where the number of samples is significantly smaller compared to pre-training—SNSA introduces minimal scalability limitations.

### A.10 Effect of SNSA on self-attention behavior on the MIMIC-IV dataset

Figure 8 shows the effect of augmenting pre-trained Transformers with NSA and SNSA on a specific sample on the MIMIC-IV dataset.

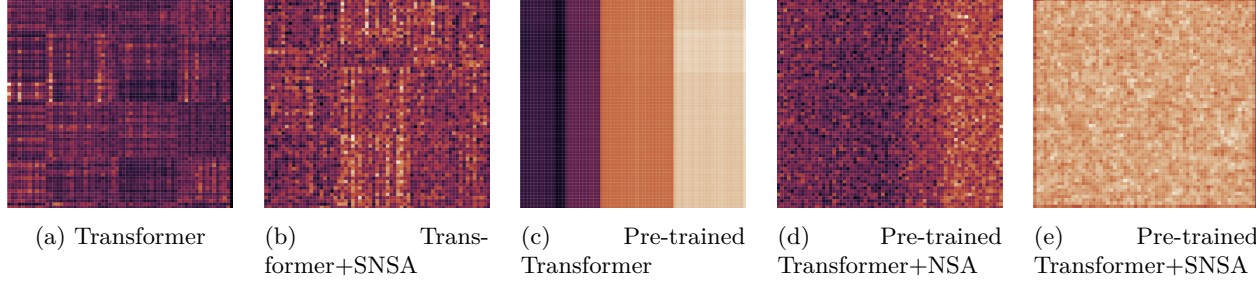

(a) Transformer    (b) Transformer+SNSA    (c) Pre-trained Transformer    (d) Pre-trained Transformer+NSA    (e) Pre-trained Transformer+SNSA

Figure 8: The attention score weights for ten fine-tuned models on HF prediction on the MIMIC-IV dataset for a specific sample.scale of the heatmaps varies across different models.

### A.11 Baseline Models

The baselines in our study were selected based on prior research and practical considerations for modeling temporal health data. The following models were used for comparison:

- **Logistic Regression (LR)**

- **Random Forest (RF)**

- **Multilayer Perceptron (MLP)**

- **Bidirectional GRU (Bi-GRU)**

- **Transformer (scratch)**: A Transformer encoder trained from scratch using multi-head attention followed by a classification feedforward head.

- **Pretrained Transformer (MLM)**: A Transformer encoder pretrained on MLM and fine-tuned on downstream tasks, following approaches such as BEHRT, Med-BERT, and others (Rasmy et al., 2021; Li et al., 2020; Meng et al., 2021).

For LR, RF, and MLP, each visit was encoded as a multi-hot vector and aggregated via summation across visits. These baselines allow us to benchmark the performance of SNSA against both classical machine learning models and modern Transformer-based architectures. Training and fine-tuning hyperparameters for each model are provided in Appendix A.1. Table 8 shows the results.

Table 8: Average AUC values (%) and standard deviation for different baseline methods for the HF prediction, AD prediction, and PLS prediction downstream tasks on the test datasets.

| Model / Downstream Task | HF prediction (MDC) | AD prediction (MDC) | HF prediction (MIMIC-IV) | PLS prediction (MIMIC-IV) |
|---|---|---|---|---|
| Logistic regression | 62.4 (1.1) | 56.4 (1.1) | 83.9 (1.2) | 54.2 (0.4) |
| Random forest | 60.7 (0.5) | 51.8 (0.3) | 77.2 (2.3) | 51.1 (0.3) |
| MLP | 67.9 (3.0) | 68.0 (1.5) | 85.2 (0.3) | 59.3 (1.9) |
| Bi-GRU | 62.3 (1.2) | 60.4 (1.1) | 86.5 (1.2) | 55.9 (1.0) |

