# OpenReview forum: "Self-Attention Augmentation with Smoothed Noise Injection for Enhancing Transformer Fine-Tuning on Temporally Structured Health Data"
_TMLR — Rejected by TMLR_

### Review · Reviewer_bqw2 · 2025-02-14

**Summary Of Contributions:**

This work investigates the problem of modeling longitudinal health data represented as a sequence of events.

The work primarily deals with the claim that vanilla transformers, despite containing an attention mechanism which has a full receptive field, after training, the attention mechanism is biased towards locality, that is, focuses primarily on relationships between events which co-occur, or approximately co-occur. The authors further claim that the attention mechanism locality harms performance, and propose two mechanisms to deal with this behavior in the fine-tuning stage:

- Global attention (where attention scores are delocalized, based on the statistics of the attention mechanism)
- Local attention (where attention are relocalized, based on an added hyperparameters $\sigma_{eh}$  and the kernel size).

The authors claim that in combination, Global-Local Attention (GLA) can fix the problems of the vanilla attention mechanism, and introduce a training algorithm that allows GLA to be implemented in a fine-tuning stage with a data augmentation process, circumventing requirements of modifying the computational graph.

Particularly when combined with masked language modeling MLM, the authors show GLA improves performances on three downstream tasks:
1. Heart Failure (HF)
2. Alzheimer Disease (AD)
3. Prolonged Length of Stay (PLS)
Treated as conditional categorical prediction tasks given the patient history up until the event to be predicted.

**Audience:**

Yes

**Broader Impact Concerns:**

No broader impact concerns.

**Claims And Evidence:**

No

**Requested Changes:**

# Required (critical)

- Motivate the work with concrete examples of model failure due to an inability to use the context properly. This can be cited out, rather than a new experiment, but it must be part of the narrative.
- Rework and link section 4.1 to the rest of the paper (see weaknesses above).
	- Specifically, provide a proof or justification that GLA induces the structure discussed in 4.1, and how this specific structure addresses issues outlined in motivation.
- Add heatmap colorbars and normalize throughout work. Explain what heatmaps are of (layer index, head index,...). For Fig 4, aggregate heatmaps over >1 sample. (see weaknesses for discussion).
- Present sensitivity analysis of hyperparameters introduced for your method.
	- Answer: are the optimal values of  $\sigma_{eh}$  and kernel size stable across your cross-validation splits?
	- How do the models perform for reasonable ranges of $\sigma_{eh}$  and kernel size? What are the reasonable ranges of these hyperparameters?
- Explain all hyperparameter choices in paper, for both MLM and fine-tuning. Be explicit about any biasing this may have on your conclusions.
- Show training and validation losses for all models. Show training downstream metrics for all models. This can be in an appendix.
- Normalize attention scores in Figure 5 (see weaknesses).
- Address the spacing in your paper, particularly, remove the additional new paragraphs surrounding each of your equations.
- Explain role of repeated SEP, or amend eq. 14 (see weaknesses).
- Include discussion of modern methods for dealing with spatio-temporal difficulties (see weaknesses).
- Use section references throughout (see weaknesses).
- Explain the pros and cons of data augmentation procedure compared with modification of the computational graph.

# Recommended for strengthening but not critical

- Include discussion regarding attention mechanism non-normalization choice.
- Use consistent table formatting. Suggest to use booktabs, see e.g. the NeurIPS table style in https://neurips.cc/Conferences/2023/PaperInformation/StyleFiles .
- Move baseline model discussion to the appendix
- Include at least one result/method from other works in Table 2.

**Strengths And Weaknesses:**

# Strengths

Modeling longitudinal health data is worthwhile and challenging to study. Findings and improvements of methods in this area have the potential to improve lives.

The authors include relevant baseline models as well as the models being modified by their proposal (although possibly these should be moved to appendix to serve readability).

The authors use cross-validation and check for statistically significant results (although which test they are using should be stated explicitly).


# Weaknesses

Overall the paper was challenging to read. If I understand, the main motivation for the proposed solution is that the attention mechanism learned by a vanilla transformer is too local, and induces problems modeling longitudinal data. To help the paper, i) this motivation needs clearly spelling out, and backing up with evidence, i.e. substantive demonstration of model failures due to a model inability to predict something sufficiently well due to an overly local bias of the attention mechanism. And ii) an explanation or discussion as to why, during standard training, if a vanilla transformer (which could in principle model the phenomena with long-range attention) chooses not to, even in the context of fine-tuning. This is particularly of interest, given the recent successes in needle-in-the-haystack type problems with attention mechanisms, and results on the Long Range Arena (https://arxiv.org/abs/2011.04006), where the context sizes are significantly larger than those studied in this work. Is there a specific challenge faced with medical data that is distinct from say, language or image data, that prevents this proper use of context?

I found 4.1 in general challenging to read, and am not entirely sure if the notation is correct, or helpful. For example, equation 5 is written as the union of sets of single elements, whereas in typical set builder notation one would write

$Atten_{L_i}=\{e_j : j\in N_i\}$

i.e. the set of event in the neighborhood of $N_i$.

Similarly then, equation 6, which is the intersection between local attentions could be written succinctly as

$Atten_{G_{i,m}} = Atten_{L_i}\cap Atten_{L_m}$.

Finally in equation 7, it is not entirely clear to me where the $i,m$ indices disappeared to on the left-hand side. Probably the union operator is being done over these indices but I cannot be entirely sure, as 4.1 in general needs more explanation, and possibly some motivating examples. Similarly, even though 4.1 introduces the concepts of Local and Global attention, the quantities developed in equations 5-7 are not mentioned in the remainder of the paper.

My expectation/guess is that in 4.2, the GLA algorithm proposed is intended to provide a training objective that would induce the attention mechanism described in 4.1 without modification of the computational graph. I.e., performing Algorithm 1 induces upon the dense matrix $A$ a structure which could be equivalently implemented by modifying the computational graph to satisfy 4.1. If this is the intention, the authors need to i) state that is the intention of GLA, and ii) prove this is the case. If this is not the intention, and I have misunderstood, the authors need to more clearly explain the role of section 4.1 and its relation to the wider work.

Heatmaps throughout the work (Figure 1, 4) need color bar scales and consistent normalization. These components are necessary in order for conclusions to be drawn for the work. Also, the authors need to state for which layers, heads, and samples these heatmaps are being computed for. Additionally, heatmaps presented are for single samples. Attention behavior can vary significantly between samples. It would be useful if the authors could aggregate the attention distributions of the methods over multiple samples, for example following Fig 8 in  https://arxiv.org/abs/2207.07611. The authors have done an aggregation in 1D similar to this in their Figure 5, which I think is helpful, and a more reliable measure.

The authors should present some indication of hyperparameter sensitivity in their work. First, the method they propose introduces two new hyperparameters. The performance sensitivity to these hyperparameters needs discussing. Second, there is no indication of the training or validation losses of the models in question in the work, so a reader has no signal for whether the result in model performance is due to 1) overfitting, 2) underfitting, 3) unfortunate choice of learning rates, or a number of other possible factors. The authors should clearly state how hyperparameter are chosen, and report the training and validation losses of the models they are using.

The authors say that a layerwise learning rate decay coefficient of 0.9 was used with an initial learning rate of 6e-5. The authors need to explain why this choice was made, and if it was optimized for any specific method in the work. Additionally, it sounds like the authors are using a learning rate schedule, they need to say which one. It also sounds like the authors are not using learning rate warmup - transformer models are notoriously difficult to train without a learning rate warmup. If this is the case, I would strongly suggest the authors try using a learning rate warmup as part of their schedule.

When the authors use a model "pretrained on MLM", what masking ratios on schedules did they use? How much data? What was the stoping criterion? Please show train and validation losses for this as well.

The resulting attention mechanism (line 6, algorithm 1) results in attention weights that are not normalized. A discussion of the implications of this would be beneficial. Additionally, this results in different y-scales across methods in Figure 5. It would be useful to either renormalize the scores for the purposes of comparison in Figure 5, or consider a version of the attention mechanism which preserves score normalization. Can the authors also comment on why the choice was made to not modify the logits with their global noise injection, analogous to their naive masking baseline (section 5.6)?

It is unclear to me what role the non-transformer baselines play in the paper as they are not mentioned in the discussion, except in a passing comment when discussing reduced data performance of GLA. I found them distracting from the main narrative, and would recommend moving the baseline models to the appendix.

There are many works that try to deal with the problems of local versus non-local attention mechanisms, and in general deficiencies of dealing with spatio-temporal information. It is unclear which positional encoding the authors are using. Assuming learnable or sinusoidal, there are many known deficiencies these encoding mechanisms lead to, which are partially alleviated by methods like ALiBi (https://arxiv.org/abs/2108.12409), and RoPE (https://arxiv.org/abs/2104.09864) which both make spatially aware modifications of the attention mechanisms. Similarly, the neighbohood attention mechanism has been successfully deployed in works like NAT (https://arxiv.org/abs/2204.07143) and LongNet (https://arxiv.org/abs/2307.02486). I would expect a discussion of these works in at least the related work section.

The main benefit the method the authors propose over NAT and LongNet the above works is that the structure of the predictions is achieved through augmentation and training, rather than modification of the model. The authors need to more strongly motivate specifically why not modifying the model is beneficial, and for example, discuss challenges there might be introducing the above methods into existing models for the purposes of fine-tuning.

The authors should include an impact statement for this work to help frame it in context of the mostly positive implications for the real world.

It is unclear in eq. 14 what the role of the SEP token has, (it always occurs every-other token, which means it contains no information about the sequence). Unless, an event is a sequence of $V_j^i$ that need separating from each-other, then such an should be shown in eq. 14.

Throughout the work, different sections and appendices are referred to in writing, but not specifically. For ease of readability, proper section references should be used. E.g. below eq 13 "see the technical appendix".

---

> ### Author Response · Authors · 2025-04-20
>
> We thank the reviewer for the thoughtful feedback. We address your comments below:
>
> Motivate the work with concrete...
>
> While GLA impacts the attention’s receptive field, its primary purpose is to enhance the quality of the learned representations in Transformer models. To clarify our contribution, we have updated the paper’s title to:
> “Self-Attention Augmentation with Smoothed Noise Injection for Enhancing Transformer Fine-Tuning on Temporally Structured Health Data.”
> Also, to better motivate our work, we revised paragraphs 4,5 and 6 of the introduction to emphasize the pre-trained Transformer model's limitations in capturing contextual dependencies in longitudinal health records. We now include specific examples and citations of cases where models fail to effectively utilize context, aligning with the reviewer's suggestion. Additionally, we updated Figure 1 and added a new Figure 4 to visually illustrate how these models struggle to learn complex event dependencies.
>
> Rework and link section 4.1, ...
>
> To address the reviewer’s concern and avoid confusion, we have removed Section 4.1 and updated the title and methodology accordingly to better reflect the core contribution of our work. Additionally, we have introduced Figure 4 and added Sections A.5, A.6, and A.7 in the appendix. These additions provide a more rigorous justification for how smoothed Gaussian noise injection enhances the model’s ability to learn richer and more generalizable representations, directly addressing the challenges outlined in our motivation.
>
> Add heatmap colorbars...
>
> We normalized all heatmaps to the [0,1] range and added colorbars for clarity. Detailed explanations have been included. Additionally, we added Figure 4 to reflect the aggregated impact of SNSA across multiple samples and all attention heads.
>
> Present sensitivity analysis...
>
> We have added a sensitivity analysis of the hyperparameters in Appendix Section A.3 and Table 6. The optimal values for σeh and the kernel size are relatively stable across cross-validation splits. Moreover, varying σeh within a reasonable range does not significantly affect model performance, indicating robustness to hyperparameter selection.
>
> Explain all hyperparameter...
>
> We have thoroughly detailed all hyperparameter choices for datasets in Appendix Section A.1. we also added pre-training and validation loss curves for all models.
>
> Normalize attention...
>
> We normalized the attention scores in fig 5. We also addressed the spacing and removed the new paragraph around equations.  We also fixed the section references.
>
> Explain role of repeated SEP...
>
> Each visit in the EHR data consists of a set of diagnoses and medications recorded within a specific time span. The SEP (separator) acts as a delimiter indicating the transition to a new visit or time period. We updated the explanation of Equation 14 to clarify this role and remove any confusion regarding its purpose
>
> Include discussion of modern methods...
>
> Thank you for the suggestion. We have updated the Related Work section to include recent methods and better motivate our approach in this context.
>
> Explain the pros and cons of data augmentation...
>
> While studies like “Equivalence between Dropout and Data Augmentation: A Mathematical Check by Zhao” et al. have shown that data augmentation can be equivalent to adding noise to a neural network model during training, data augmentation is not always straightforward. Longitudinal EHRs are represented in a discrete input space (e.g., medical codes or text), and augmenting such data is more challenging compared to continuous input spaces (e.g., images), where data augmentation can be done by adding noise or cropping or rotating and … .
> In methods like SNSA (perviousely GLA), noise is added to the latent representations of the input. By introducing different noise into the latent space, the model learns multiple views of the data in each batch, effectively simulating an ensemble of models. However, a key question is how and where in the model this on-the-fly augmentation should be applied. Noise can be injected at various stages of the model, at the self-attention layers, or in the final dense layer.
> Once the optimal layer is identified, the next consideration is how much noise to apply. While methods like Drop-atten propose masking some of the attention heads completely, SNSA shows that such an approach may lead to significant information loss. To mitigate this, adjusting the values of attention scores, rather than masking them entirely, can help preserve information while still introducing the necessary noise for augmentation.We reflect this point on the introduction and the related works section.

---

### Review · Reviewer_x27b · 2025-02-20

**Summary Of Contributions:**

This paper introduces GLA, a simple two-step augmentation method that enhances self-attention during fine-tuning without modifying the computational graph. This makes it compatible with any pretrained Transformer model. GLA captures both long- and short-range dependencies by:
- Adding noise to the self-attention matrix, encouraging attention to distant events (global attention).
- Applying a 2D Gaussian convolution, promoting attention to local events.

Key contributions can be summarized:
- Propose a lightweight augmentation method for self-attention that improves both global and local attention during fine-tuning.
- Conduct extensive evaluations on multiple downstream tasks, demonstrating improved performance, robustness in low-data settings, and a balanced attention distribution.
- Show that GLA effectively enhances pretrained Transformers without architectural modifications.

**Audience:**

No

**Broader Impact Concerns:**

No ethical implications is observed.

**Claims And Evidence:**

No

**Requested Changes:**

I have summarized the major issues in the Weaknesses section. It would be expected for the authors to address these major concerns explicitly to clarify how they have improved the manuscript. Further questions and suggestions are below.

Q1. Figure 1 is difficult to understand. The authors mention that Figure 1(a) shows attention score patterns from the vanilla Transformer and argue that it demonstrates a poor structure. However, this reasoning is unclear, as the figure lacks axis labels and contextual information. A clearer and more reader-friendly description would help improve comprehension.

Q2. Why is global attention defined as the interaction between local attentions? Intuitively, global attention should capture long-range dependencies beyond the scope of local attention. If it is constructed by combining local attention mechanisms, it would be helpful to clarify how this composition effectively models global interactions and whether any information loss occurs in the process. A more detailed explanation or justification would improve understanding.

Q3. Are the AUC values in Figure 3 consistent with those in Table 1? Table 1 reports MLM+GLA as 87.2 (0.4) and MLM as 85.2 (1.1), but the values in the plot appear to be different. Could you clarify this?

Q4. The trend in Figure 3(a) and (b) appears inconsistent. In Figure 3(a), the gap between MLM+GLA and MLM decreases as the training sample size increases. However, in Figure 3(b), the gap widens instead, which makes it difficult to interpret the results. Could you provide an explanation for this opposite trend?

Polishing the manuscript is necessary, as there are errors in capitalization. For example:
- "However, Adapting …" (Incorrect capitalization of "Adapting").
- "Where, WO∈R∣h∣×dvW^O \in \mathbb{R}^{|h| \times d_v}WO∈R∣h∣×dv​ Multiple …" (Unnecessary capitalization of "Multiple").
- In Table 2, “transformer pre-trained on” -> “Transformer pre-trained on”

Recommendation)

If the authors aim to propose a novel opt-in method for enhancing the representation of a pre-trained Transformer during the fine-tuning phase, it is crucial to review more related work and compare your approach with existing methods to demonstrate its effectiveness.
If that is not the goal, then the authors would consider incorporating characteristics of the downstream task (i.e., temporally structured health data) into your module. While this may limit the generalizability of your method, it could be more valuable for researchers working specifically in this domain. In its current form, the manuscript lacks clarity on which audience would benefit from this work, making it difficult to assess its impact. Therefore, I am inclined to recommend rejection.

**Strengths And Weaknesses:**

**Strengths**

GLA is simple and easy to understand. It does not alter the architecture of the pretrained network and requires only a few additional parameters, making it highly useful for fine-tuning in low-data settings.

**Weaknesses**

My major concerns of this work is limited novelty and insufficient experiments.

GLA consists of two key components: adding Gaussian noise to the attention matrix and applying a convolution operator to the noise-added attention matrix to facilitate better mixing of local information. However, adding Gaussian noise to activations is a well-established technique in Bayesian deep learning, commonly used to improve representation learning in low-data regimes. For example, similar approaches have been explored in:
- Variational Dropout and the Local Reparameterization Trick, https://arxiv.org/abs/1506.02557, NeurIPS’15.
- Explicit Regularisation in Gaussian Noise Injections, https://arxiv.org/abs/2007.07368, NeurIPS’20.

One of the key motivations behind GLA is to fine-tune a pretrained Transformer with minimal changes, particularly in low-data settings. However, there are well-established approaches for this purpose, collectively known as Parameter-Efficient Fine-Tuning (PEFT). A representative method in PEFT is LoRA (Low-Rank Adaptation), which has been widely studied and adopted in various domains. Given this context, GLA should be compared to prominent PEFT methods to justify its necessity and effectiveness. Otherwise, it remains unclear why researchers should adopt GLA over existing PEFT techniques for their applications.

Additionally, the performance improvement appears to be marginal and may not be statistically significant. In Table 1, the confidence intervals of the results for MLM and MLM+GLA seem to overlap, particularly in HF prediction (MDC), AD prediction (MDC), and PLS prediction (MIMIC-IV). This suggests that the observed improvements may not be meaningful.

Finally, the overall quality of the manuscript is not high enough to meet the acceptance standards of TMLR. I recommend revising and polishing the manuscript to improve clarity, coherence, and technical rigor before resubmission.

---

> ### Author Response · Authors · 2025-04-20
>
> We thank the reviewer for the thoughtful feedback. We address your comments below:
>
> Figure 1 is difficult...
>
> (Q1, motivation and novelty:) While GLA (now renamed to SNSA) affects the attention’s receptive field, its primary goal is to enhance the quality of the learned representations in Transformer models during fine-tuning. To clarify our contribution and better reflect the method's intent, we have updated the title of the paper to:“Self-Attention Augmentation with Smoothed Noise Injection for Enhancing Transformer Fine-Tuning on Temporally Structured Health Data.”We revised the introduction, related work, and Figure 1, and added Figure 4 to strengthen the motivation and clarify the novelty of our method.
>
> We appreciate the references suggested. While prior have shown the value of adding noise to activations, they do not explore where and how noise should be injected in pre-trained Transformer architectures. Our proposed method, SNSA, focuses specifically on injecting adaptive Gaussian noise into the self-attention scores—a crucial component for modeling dependencies in sequential data. Moreover, this is followed by a Gaussian smoothing step that controls the perturbation and promotes more meaningful attention structures. This combination is not explored in the cited works.
>
> Why is global attention...
>
> While GLA (now referred to as SNSA) does affect the attention’s receptive field, its primary goal is to improve the quality of the learned representations in Transformer models. To clarify our contribution, we have updated the paper’s title, as well as the introduction and related works sections.
> Additionally, to provide stronger justification for how SNSA enhances representation learning, we have extracted Figure 4 and added Appendix Sections A.6, A.7, and A.8, which offer both mathematical analysis and visualizations supporting the benefits of our approach.
>
> Are the AUC values...
>
> The AUC values reported in Figure 3 are consistent with those in Table 1. To improve readability and avoid confusion, we have added grid lines to the plots in Figure 3, making the numerical values easier to interpret and align visually with the table.
>
> The trend in Figure 3...
>
> The performance improvement from augmenting self-attention (SNSA) is typically in the range of 2–3%. However, in the MDC dataset—especially when using only 10% of the data—the model's predictions are less stable due to limited training samples and is around random. Additionally, the difference in y-axis scaling between Figure 3(a) and 3(b) may have contributed to the visual inconsistency. To address this, we provide the full numerical results in Table 7 (Appendix), which show that the improvements remain consistent across sample sizes, despite the apparent divergence in the plotted curves.
>
> Polishing the...
>
> We polished the manuscript and fixed the mention issues.

---

### Review · Reviewer_m2kG · 2025-03-29

**Summary Of Contributions:**

1. Adding noise Adaptively scales with the mean and variance of the attention matrix. This is to foster “global exploration” of distant tokens/events (long-range attention).

2. Gaussian Smoothing removes some added noise, and encourages each token to look at its neighborhood in a controlled radius, aiming to foster local structure.

3. Since the method is purely an on-the-fly modification of the attention matrix, the standard transformer structure remains intact.

**Audience:**

Yes

**Broader Impact Concerns:**

NAN

**Claims And Evidence:**

No

**Requested Changes:**

See above.

**Strengths And Weaknesses:**

Weakness:
1. GLA involves adding an adaptive Gaussian noise directly to the attention matrix. While this is presented as a heuristic for enhancing both local and global attention, the explanation is mostly intuitive. No strong theoretical derivation shows why injecting noise into the attention scores is definitely the best way to capture long-range dependencies.
By comparison, existing regularization/noise methods—such as dropout, random masking, or adversarial training—often have broader empirical or theoretical support. GLA’s justification relies more on anecdotal reasoning and visualization.

2. Gaussian convolution is commonly used in image processing to remove noise. smooth pixel intensities. Transplanting it onto the attention matrix, which is a “token-by-token” or “event-by-event” matrix, may not be inherently well-justified in discrete domains like NLP or medical coding. And why a 2D Gaussian kernel? Why not a different kernel shape or a more adaptive weighting? There is no universal argument provided. Adding noise then removing it sounds lacking sense.

3. A Transformer’s attention matrix is presumably the model’s best learned representation of dependencies. By injecting noise and convolving, GLA intentionally distorts that distribution.  If the model has found a good local/global pattern on its own, GLA’s additional smearing and randomization may regress performance or require extra epochs to repair the unwanted perturbations.

4. The key hyperparameter \(\sigma_{eh}\) determines the extent of the Gaussian neighborhood. A large kernel may smooth attention so aggressively that local specificity is lost; a small kernel may fail to spread any meaningful local structure. Determining the ideal value for each new dataset or task can be cumbersome.  The same challenge applies to the noise variance \(\sigma_{GN}^2\), which depends on how wide or narrow the attention distribution is. Overly large noise can disrupt beneficial patterns; insufficient noise might not achieve the intended goal of exploring far tokens.

5. Modern transformers and fine-tuning pipelines often include dropout, weight decay, gradient clipping, or adversarial training. GLA is an additional form of perturbation that might conflict with or duplicate effects of other techniques.

6. During inference, the method replaces the noise with its mean \(\mu\). This is a somewhat ad-hoc choice—there are other ways to handle noise injection at test time (e.g., sampling multiple times, or removing noise entirely). GLA’s exact approach may not generalize equally well to all tasks or domains.

7. Criticism: The HF results show GLA improves performance by 3%–4% in certain settings. However, the authors do not mention if hyperparameters (kernel size) were re-tuned for each data fraction. Additional ablation clarifying how sensitive GLA is to these hyperparameters in data-scarce environments would strengthen the claims.

---

> ### Author Response · Authors · 2025-04-20
>
> We thank the reviewer for the thoughtful feedback. We address your comments below:
>
> Q1:
>
> While GLA (now referred to as SNSA) does affect the attention’s receptive field, its primary goal is to improve the quality of the learned representations in Transformer models. To clarify our contribution, we have updated the paper’s title, as well as the introduction and related works sections.
> Additionally, to provide stronger justification for how SNSA enhances representation learning, we have extracted Figure 4 and added Appendix Sections A.6, A.7, and A.8, which offer both mathematical analysis and visualizations supporting the benefits of our approach.
>
> Q2:
>
> We agree it requires justification. Our motivation for applying a 2D Gaussian kernel stems from the structure of the attention matrix itself, which represents pairwise interactions between tokens or events. Each element Ah(i,j) captures the attention from token i to token j, forming a two-dimensional space of interactions. Therefore, the attention map is inherently 2D, justifying the use of a 2D kernel over 1D alternatives.
>
> The Gaussian kernel serves a specific role: rather than simply denoising, it performs controlled smoothing. After injecting adaptive Gaussian noise into the attention scores, the convolution acts as a low-pass filter, reducing high-frequency fluctuations introduced by the noise. This preserves meaningful patterns while encouraging the model to generalize beyond sharp, binary-like decisions (as we observed in Fig. 4 and Appendix A.6).
> We do not "remove" the noise in the traditional sense, but rather redistribute and regularize it. The convolution adjusts the magnitude and locality of the injected noise, enabling the model to focus on broader, smoother interactions across the attention map. The event horizon parameter σeh\sigma_{eh}σeh offers tunable control over how much locality is preserved.
> Regarding the choice of kernel: we used Gaussian smoothing as it is well-understood, easy to parameterize, and introduces minimal inductive bias. We have clarified this motivation in the updated methods and appendix.
>
> Q3:
>
> Indeed, the attention matrix reflects the model’s current understanding of inter-token dependencies, and introducing any perturbation must be done carefully. Our method, Smoothed Noise Injection Self-Attention (SNSA), does not aim to distort well-learned attention patterns but rather to encourage exploration of alternative dependencies during fine-tuning—especially in low-data or high-sparsity settings, which are common in temporally structured healthcare data. Without such augmentation, we observed (as shown in Fig. 4 and Appendix A.6) that attention distributions often become overly binary, locking into local minima that miss complex global interactions.
> The injected Gaussian noise is adaptive, based on the existing attention score statistics (mean and variance), ensuring that perturbations are scaled appropriately to the model’s confidence. The subsequent Gaussian smoothing further regularizes this noise, promoting coherent local structure instead of random distortion. We reflected it in the revised introduction and related works.
>
> Q4:
>
> The mean and standard deviation of the added Gaussian noise are adaptively computed per attention head, based on the actual attention distribution in each forward pass. This dynamic strategy eliminates the need to manually tune σGN, making the noise adaptive to the model’s own confidence in its attention patterns. The Gaussian smoothing kernel, σ_eh is a single tunable hyperparameter. However, our sensitivity analysis in Appendix A.3 (Table 6) shows that the model is robust to a wide range of σ_ehσ values. While extremely small or large values can slightly degrade performance.
>
> Q5:
>
> While it is true that modern training pipelines include several regularization and stabilization techniques, SNSA complements rather than duplicates their effects (similar approaches are discussed in the related works sections). Dropout and weight decay operate on model weights or activations and aim to prevent co-adaptation and overfitting globally. In contrast, SNSA operates specifically on the attention score matrix, targeting the model’s inductive bias toward hard and narrow attention patterns. We compared SNSA with DropAttention and naive self-attention masking methods from the literature and demonstrated that SNSA leads to more effective and stable performance (see Table 3 and Section A.7). Furthermore, as illustrated in Figure 4, SNSA encourages the model to learn more diverse and robust attention distributions, ultimately improving its ability to capture dependencies during fine-tuning.

---

> > ### Author Response · Authors · 2025-04-20
> >
> > Q6:
> >
> > We acknowledge that replacing the noise with its mean μ during inference is a design choice that trades off simplicity and consistency. This approach ensures deterministic behavior at test time, which is critical in clinical applications where reproducibility and stability are essential. While other approaches—such as Monte Carlo sampling or removing the noise entirely—are valid, they introduce variability or risk losing the benefits gained during training. We chose the mean substitution to retain the bias induced during training, without introducing additional stochasticity or inference-time complexity. Importantly, our empirical results demonstrate that this method performs robustly across tasks and datasets.
> >
> > Q7:
> >
> > We have addressed the sensitivity of hyperparameters in detail in Appendix Sections A.1.2 and A.3. In particular, Section A.3 presents a sensitivity analysis showing that variations in the Gaussian smoothing parameter σeh and corresponding kernel size do not significantly impact model performance. We also observed that the optimal value of σeh remains consistent across tasks within the same dataset, suggesting robustness of the method to this hyperparameter.

---

### Decision · Action_Editor_N88Z · 2025-06-04

**Recommendation:** Reject

**Additional Comments:**

NA

**Audience:**

Yes

**Audience Explanation:**

The claim about the effectiveness of the proposed Self-Attention Augmentation with Smoothed Noise Injection can strengthened by adding comparison with existing fine-tuning baselines.
Reviewer m2KG mentioned that "*two critical gaps remain: The method is not compared to canonical parameter-efficient fine-tuning baselines (LoRA, Bit-Fit, AdapterFusion). ... the theoretical section is illustrative rather than principled.*" Reviewer x27b remarked that "*the rebuttal did not adequately address my question regarding comparisons with PEFT approaches. These methods are widely used and share the same motivation as this work, fine-tuning networks with minimal architectural modification*".

The paper proposes to add Gaussian noise to the attention layer of Transformer with a goal to improve the fine-tuning on downstream tasks on medical data. Since Transformer is a widely used model, it will have interest from TMLR audience.

**Claims And Evidence:**

No

**Claims Explanation:**

After the authors' responses, several reviewers claim that their concerns are not fully addressed, including

Missing Baselines: No comparison with standard parameter-efficient fine-tuning methods (e.g., LoRA, BitFit), which undermines the empirical claims. This  has not been well addressed in the responses and revision.

Weak Theoretical Justification: The theoretical section is illustrative, not rigorous or clearly linked to the method.

Based on these comments, I will have to recommend rejection and encourage the authors to revise it and submit a a major revision at a  later time.

**Resubmission Of Major Revision:**

The authors may consider submitting a major revision at a later time.